# Linking Biogenic High-Temperature Ice Nucleating Particles in Arctic soils and Streams to Their Microbial Producers

Lasse Z. Jensen[1,2,3a*], Julie K. Simonsen[1,4a], Ada Pastor[4,7], Christof Pearce[2,3,6], Per Nørnberg[1,6†], Lars Chresten Lund-Hansen[2,4], Kai Finster[1,2,3,5], Tina Šantl-Temkiv[1,2,3,5*]

[1]Department of Biology, Microbiology Section, Aarhus University, Ny Munkegade 116, 8000 Aarhus, Denmark.

[2]Arctic Research Center, Aarhus University, Ny Munkegade 116, 8000

Aarhus, Denmark.

[3]iCLIMATE Aarhus University Interdisciplinary Centre for Climate

Change, Frederiksborgvej 399, Roskilde, Denmark.

[4]Department of Biology, Aquatic Biology Section, Aarhus University, 8000 Aarhus C, Denmark.

[5]Stellar Astrophysics Centre, Department of Physics and Astronomy, Aarhus University, Ny Munkegade 120, 8000 Aarhus, Denmark

[6]Department of Geoscience, Aarhus University, 800 Aarhus C, Denmark.

[7]Group of Continental Aquatic Ecology Research (GRECO), Institute of Aquatic Ecology,

University of Girona, Girona, Spain

[a]These authors contributed equally to this study.

[†]Author deceased

*Correspondence to*: Tina Šantl-Temkiv (Temkiv@bio.au.dk) and Lasse Z. Jensen (Lassejensen@bio.au.dk)

**Abstract.** Aerosols, including biological aerosols, exert a significant influence on cloud formation, influencing the global climate through their effects on radiative balance and precipitation. The Arctic region features persistent mixed-phase clouds, which are impacted by ice nucleating particles (INPs) that modulate the phase transitions within clouds, affecting their lifetime and impacting the region's climate. An increasing number of studies document that Arctic soils harbour numerous biogenic INPs (bioINPs), but these have yet to be linked to their microbial producers. In addition, the transfer of bioINPs from soils into freshwater and marine systems has not been quantified. This study aimed at addressing these open questions by analyzing soil and freshwater samples from Northeast Greenland to determine the microbial composition along with the INP concentrations and size distributions. We found that soils contained between $3.19 \cdot 10^4$ and $1.55 \cdot 10^6$ INP $g^{-1}$ soil, which was on the lower side of what has previously been reported for active layer soils. The composition of INPs varied widely across locations and could have originated from bacterial and fungal sources. We detected *Mortierella*, a fungal genus known to produce ice-nucleating proteins, in nearly all locations. Spearman correlations between soil taxa and INP concentrations

pointed at lichenized fungi as a possible contributor to soil INP. Additionally, based on the INP size distribution, we suggest that soil INPs were bound to soil particles or microbial membranes at some locations, while other locations showed a variety of soluble INPs with different molecular sizes. In streams, INP concentrations and onset temperatures were comparable to

what has previously been measured in streams from temperate regions. Interestingly, stream INP concentrations showed a positive association with soil INP concentrations. The potential release and aerosolization of these bioINPs into the atmosphere - whether directly from the soil, from streams into which they are washed, or from the oceans where they might be transported - could impact cloud formation and precipitation patterns in the Arctic. This research contributes valuable knowledge to the understanding of microbial communities and the potential producers of highly active bioINPs in Arctic soil microbial

communities and their connectivity with Arctic streams.

## 1 Introduction

Clouds affect the global climate by regulating both surface precipitation and the atmosphere's radiative balance by absorbing, reflecting, and scattering radiation (Fan et al., 2016; Huang et al., 2021). Atmospheric aerosols affect cloud properties and precipitation through aerosol-cloud interactions, by acting as either cloud condensation nuclei to form cloud droplets or as ice

nucleating particles to form ice particles (Fan et al., 2016; Möhler et al., 2007). Among the various aerosol types, primary biological aerosol particles (PBAP ), have been recognized already ~50 years ago for their high activity and wide distribution (Schnell and Vali, 1976; Vali et al., 1976). Recently, they have regained broader attention due to their potential to impact cloud formation and climate (Huang et al., 2021).

According to the Intergovernmental Panel on Climate Change (IPCC) report from 2023, the largest uncertainties in estimates

of the Earth's changing energy budget are associated with cloud adjustments due to aerosols (Ipcc, 2023). The Arctic region is especially sensitive to factors that trigger climate change due to Arctic amplification, which is the phenomenon that greenhouse-induced warming is enhanced and accelerated in the Arctic due to positive feedbacks between atmospheric processes, such as reduced convection and intrusion of humid air masses from the equatorial regions, and reduced extent of sea ice, terrestrial ice, and snow cover (Holland and Bitz, 2003). Arctic amplification is responsible for a 2.5 °C higher surface

air temperature increase in the Arctic compared to the global average (Serreze and Francis, 2006; Serreze and Barry, 2011). The predominant cloud type in the Arctic is mixed-phase clouds, which are composed of a mixture of supercooled droplets and ice particles (Fan et al., 2016). Arctic mixed-phase clouds are especially long-lived and can persist for several days (Morrison et al., 2011; Fan et al., 2016). They are important factors in controlling the Arctic climate, as they have a large impact on surface radiative fluxes and thus the energy balance (Fan et al., 2016; Morrison et al., 2011), which affects the ice-

melting rate (Carrió et al., 2005).

The number concentration of ice particles in mixed-phase clouds is especially important because it influences the content of supercooled droplets, the size distribution of cloud particles, and hence, the optical properties of the cloud. Ice formation in the atmosphere can occur in two ways, by either homogenous or heterogeneous ice nucleation. Homogeneous freezing takes

place in droplets of pure water that freeze at temperatures below -38 °C (Bigg, 1953). Heterogeneous freezing requires the presence of nuclei such as dust that facilitate the ice formation. The particles are called ice nucleating particles (INPs). In their presence water droplets freeze at temperatures above -38°C (Kanji et al., 2017). This is the case in mixed-phase clouds, where INPs greatly affect the lifetime, and the radiative properties of the clouds (Hartmann et al., 2019).

INPs stem from various sources e.g., soil, vegetation, marine sea spray, and freshwater and can enter the atmosphere as, e.g., emissions of desert dust, sea-spray particles, anthropogenic emissions, and emissions from (micro)organisms (Huang et al., 2021).

Ice nucleation below -13°C can be initiated by abiotic INPs, such as mineral particles, soot, or by incidental ice nucleation activity from biomolecules, such as carbohydrates (Kinney et al., 2024). In contrast, proteinaceous INPs are the predominant type of INP that are active above -13°C, shown by both laboratory studies and in-situ measurements involving inactivation by heating (Murray et al., 2012; Cornwell et al., 2023; Kanji et al., 2017; Daily et al., 2022). Biological INPs include bacterial, fungal, algal cells, as well as cell fragments and macromolecules and have all been found in the Arctic atmosphere (Santl-Temkiv et al., 2022; Jensen et al., 2022; Santl-Temkiv et al., 2019; Kanji et al., 2017; Huang et al., 2021; Bigg, 1996; Bigg and Leck, 2001; Creamean et al., 2022; Stopelli et al., 2015; Stopelli et al., 2017; Hartmann et al., 2020; Hartmann et al., 2021; Ickes et al., 2020; Jayaweera and Flanagan, 1981; Porter et al., 2022).

While some recent studies focused on the sources and abundance of ice-nucleation active (INA) bioINPs in the Arctic (Santl-Temkiv et al., 2019; Pereira Freitas et al., 2023; Sze et al., 2023), there remains a significant knowledge gap when it comes to determining their types, quantifying their numbers, and identifying their producers in the soil environment. Still, little is known about microbial producers responsible for the production of bioINPs in active layer soils as well as their transfer to freshwater and marine systems. Studies focusing on temperate regions, have found that soil contained highly active INPs (Conen et al., 2011; Hill et al., 2016; O'sullivan et al., 2016). These INPs have been associated with soil fungi affiliated to the genera *Fusarium* and *Mortierella*, which can produce ice-nucleating proteins (INpro) (Fröhlich-Nowoisky et al., 2015; Pouleur et al., 1992). Highly active INPs have also been found in rivers in temperate regions (Larsen et al., 2017; Knackstedt et al., 2018; Moffett, 2016; Moffett et al., 2018), which, once aerosolized, may impact regional clouds, and thus the climate. Similar to the emissions from seawater, aerosols can be formed from rivers and streams associated with turbulence due to high velocity and slopes, rain splash or obstacles interfering with the flow of water (Huang et al., 2021; Knackstedt et al., 2018; Raymond et al., 2012). Recent studies focusing on the Arctic found highly active INPs in active layer and thawing permafrost soils and linked them to watersheds through wash-out and melting processes (Creamean et al., 2020; Barry et al., 2023a; Barry et al., 2023b; Tobo et al., 2019). While the direct quantification of aerosolization is beyond the scope of this study, understanding the transport and potential atmospheric impact of bioINPs from Arctic active layer soils and freshwater systems is crucial. The pathways through which these particles may enter the atmosphere can significantly influence cloud microphysics and climate dynamics

This study aims to investigate the following hypotheses: firstly, that specific microbial communities in terrestrial Arctic soils are a significant source of bioINPs, and secondly, that there is a measurable link between soil and freshwater systems regarding the transport of these INPs in Northeast Greenland. By analyzing the composition of the microbial communities in soils and quantifying INP concentrations and their size distribution in both soil and freshwater samples from adjacent streams, we aim to identify potential fungal and bacterial producers of INPs and quantify the outwash of INPs from soil into freshwater environments. This approach will allow us to estimate the overall transport and cycling of INPs in Arctic ecosystems, providing data to reduce uncertainties concerning cloud microphysics, climate feedbacks, and cloud dynamics.

## 2 Methods

### 2.1 Sampling

The streams that were studied are located in the Northeast Greenland National Park, near the Zackenberg Research Station (74°28'N, 20°34'W). Streamflow in the region is predominantly derived from melting snow and glaciers, with additional contributions varying by stream. For example, Kærelv A, Kærelv C, Grænseelv, and West 1 receive water from small, seasonal snow patches, while Aucella, and Jurassic1 is partly fed by larger ice aprons adhered to mountainsides (Docherty et al., 2019; Hasholt and Hagedorn, 2000). This region has a polar tundra climate and is underlain by continuous permafrost with an active layer thickness of 0.4 - 0.8 m (Christiansen et al., 2008; Hollesen et al., 2011). Geologically, the area is divided into crystalline complexes to the west and sedimentary successions to the east, with Quaternary sediments covering the valley floor and slopes. For a broader overview of the region's climate, geology, and vegetation, see Riis et al. (2023). Surface soil and adjacent freshwater samples from streams were collected at 12 different locations in Zackenberg, Greenland (Fig. 1 and Table 1). The samples were collected in September 2021 over a course of 11 days. From each location, 14 mL of stream water and 14 mL of filtered stream water (0.22 μm PES) was collected for INP quantification. From each location, approximately 3 g of surface soil was collected using a sterile spoon for INP quantification and microbial community profiling, making sure to collect the most abundant soil type without vegetation cover. No soil was collected at Jurassic 1, because the soil was frozen. All samples were stored at -20°C. Freshwater and soil samples were also collected for chemical and physical characterization, which was reported by Riis et al. (2023). Water chemistry parameters were measured and snow cover percentage for each stream catchment were estimated by satellite images (Riis et al., 2023).

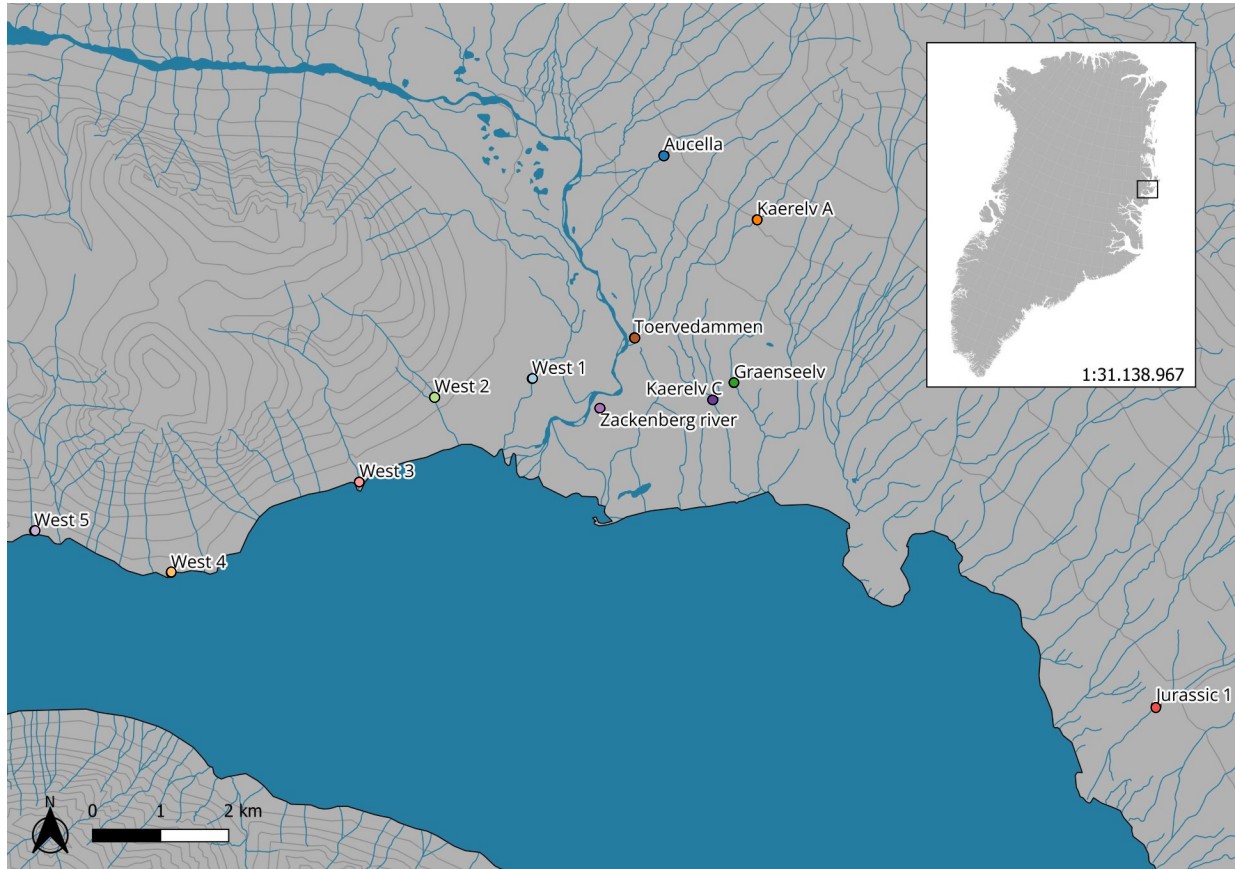

**Figure 1** Map showing the locations of the twelve sampling sites in Zackenberg, Greenland. The map was created using QGIS. Geographical data from Dataforsyningen (dataforsyningen.dk) with the Google Maps terrain as background © Google Maps.

## 2.2 Quantification of INP

We used the Micro-PINGUIN setup to determine INP concentrations as a function of temperature (Wieber et al., 2024a). The Micro-PINGUIN consists of a cooling base, where a 384-well PCR plate is inserted, and a camera tower equipped with a thermal camera (FLIR A655sc) that measures the temperature in individual wells and allows for the determination of nucleation events. The tower was first flushed with HEPA filtered clean air for 5 minutes at a constant flow of 20 L/min before. Then, the samples were cooled at 1K min$^{-1}$ down to -30°C while supplying a constant HEPA filtered clean airflow between 5-10 L/min to keep the relative humidity low, avoiding condensation.

For each of the non-filtered freshwater samples, three dilutions were made (1:10, 1:100, and 1:1000). We sequentially filtered selected samples through 1000, 300 and 100 kDa (Vivaspin®, Sartorius). The freshwater samples, the dilutions, and the filtered freshwater samples were analyzed using the Micro-PINGUIN setup. Eighty 30 µL droplets were analyzed per sample. Filtered Milli-Q (0.1 µm) was used as a negative control. A custom-made software (Ice Nucleation Controller) was used to obtain the frozen fractions.

The mean number of INPs per volume in the sample that are active at a given temperature can be calculated assuming that the locations of these INPs in the sample volume are statistically independent. Then, the probability of having a given number of droplets without INPs (that is, the fraction of no frozen droplets) is given by the binomial distribution, and the remaining fraction of droplets containing INPs (fraction of frozen droplets) leads to the following general Eq. (1):

$$N_v = \frac{-\ln(1-f)}{V \cdot \alpha \cdot \ln\left(\frac{\alpha}{\alpha-1}\right)}, \tag{1}$$

Where $f$ is the frozen fraction (i.e., the fraction of frozen droplets) and $V$ is the droplet volume in each vial of the assay and $\alpha$ is the number of droplets per sample (80 in this study).

Given that $\alpha \gg 1$ in our study, the term $\alpha \cdot \ln\left(\frac{\alpha}{\alpha-1}\right)$ is close to unity. Therefore Eq. (1) simplifies to Eq (2)

$$N_v = \frac{-\ln(1-f)}{V}, \tag{2}$$

Subsequently this simplified form was used for the calculations in this study (Vali, 1971).

We prepared soil samples for ice nucleation analysis as previously described, with slight modifications (Conen and Yakutin, 2018). The soil samples were placed in a small petri dish and freeze dried overnight (Edwards Micro Modulyo Freeze Dryer). Freeze-drying was chosen instead of air-drying to minimize the potential for microbial activity during the drying process and to preserve the original composition of INPs. Prolonged air-drying could allow for microbial activity while there is still sufficient liquid water available, which may alter INP concentrations due to production or degradation. The freeze-dried samples were kept in a desiccator to prevent rehydration and subsequently comminuted in a mortar by hand. Mortaring was performed to break down aggregates formed during freeze-drying and ensured effective sieving. Samples were dry sieved with a 125 μm and 63 μm sieve for two minutes using a vibratory sieve shaker (Analysette 3 PRO, Fritsch). The <63 μm fraction was collected for analysis, as this size range represents particles most likely to aerosolize (Fröhlich-Nowoisky et al., 2016). Hundred mg of dry <63 $\mu m$ soil particles was weighed into an Eppendorf tube. For many samples, there was less than 0.1 g soil after sieving. Instead, all the sieved soil was added to the Eppendorf tube and the weight was noted. 1 mL filtered Milli-Q (0.22 μm PES) was added to the Eppendorf tube, then vortexed for two minutes and afterwards allowed to settle for 10 minutes. 0.5 mL was withdrawn from the top of the suspension and added to a falcon tube with 9.5 mL of filtered Milli-Q (0.22 μm) creating a 1:20 dilution.

The soil suspensions were analyzed using the ice nucleation assay as described above. The number of INPs per gram that were active at a given temperature was calculated using the Eq. (3):

$$N_m = \frac{-\ln(1-f)}{V} \cdot \frac{V_s}{m}, \tag{3}$$

Where $f$ is the frozen fraction, $V$ is the droplet volume in each vial of the assay, $V_s$ is the volume of the suspension, and $m$ is the mass of soil in the suspension (Vali, 1971). Given the similar derivation of Eq. (3) to Eqs. (1) and (2), the assumption $\alpha \gg 1$ applies here as well.

Freezing onset temperatures were calculated as the temperature where 5% of the wells for a given sample had frozen. $T_{50}$ values were calculated as the temperature where 50% of all wells for a given sample had frozen.

Additionally, we extracted the concentration of ice nucleation active (INA) particles at -10°C ($INP_{-10}$) and -15°C ($INP_{-15}$) per gram of soil or mL of stream water, for use in correlation analysis.

### 2.3 Soil Total Carbon and Nitrogen measurements

Total carbon (TC) and Total Nitrogen (TN) was determined by combusting dry ball-mill-powdered soil in an elemental analyzer (Anca GSL2, Serco), coupled to an isotope ratio mass spectrometer (Hydra 20-22, Sercon). Aliquots between 10.06 and 31.83 mg soil were packed into tin cups and burned in the elemental analyzer. The content of carbon and nitrogen is given as mg C or N per kg dry soil.

### 2.4 Statistical analyses

The correlation between INP concentrations in water and soil were tested using Spearman's rank correlation analysis. The correlation between INP concentrations, water chemistry, total carbon, total nitrogen parameters were also tested using Spearman's rank correlation analysis. Significant difference in $T_{50}$ values were computed by pooling all samples into different treatment groups and conducting a Kruskal-Wallis test. Significant differences between specific treatment groups where then investigated using a post-hoc Wilcoxon rank-sum test.

The correlation analyses were done in RStudio version 4.3.0 using Vegan (Dixon, 2003).

### 2.5 Grain Size Analysis

The particle size distributions were measured at the department of Geoscience of Aarhus University on a Sympatec Helos laser diffraction instrument equipped with a wet dispersion input chamber. Samples were analyzed on R1 and R4 lenses and measurements were subsequently combined to cover the range from 0.1 – 350 μm (Rasmussen, 2020). Only four samples contained enough particles for the analysis. The four samples which were analyzed were obtained from different geographical regions of the study area.

## 2.6 DNA Extraction, Quantitative Polymerase Chain Reaction (qPCR) and amplicon sequencing

Amplicon sequencing targeting the 16S rRNA as well as the Internal transcribed spacer (ITS) region was used to determine the composition of the bacterial and fungal communities, respectively, while qPCR targeting the 16S rRNA gene was used to determine the quantity of bacteria in the soil.

DNA was extracted from 0.22-0.29 g of soil from samples where enough soil was present (Supplementary Table 1) following the power soil pro kit protocol (Qiagen). Quantitative Polymerase Chain Reaction (qPCR) using an MX3005p qPCR instrument (Agilent, Santa Clara, CA, United States) was performed to quantify the amount of bacterial 16S rRNA gene copies. We targeted partial 16S rRNA gene sequence using universal primers Bac908F (5′-AAC TCA AAK GAA TTG ACG GG-3′) and Bac1075R (5′-CAC GAG CTG ACG ACA RCC-3′) (Ohkuma and Kudo, 1998). The qPCR mixture contained 2 μl template DNA, 1 μl of each primer (10 pmol μl$^{-1}$), 2 μl Bovine serum albumin (BSA) (10 ng μl$^{-1}$), 10 μl LightCycler Mastermix (Roche) and 4 μl dH$_2$O to a final volume of 20 μl. Samples were run in triplicate including triplicate negative qPCR controls without addition of template DNA. Samples were run on a Stratagene Mx3005P (Agilent technologies) with the following thermal conditions: 95 °C for 5 min, 45 cycles with denaturation at 95 °C for 30 s, annealing at 55 °C for 30 s, elongation at 72 °C for 15 s, followed by fluorescent acquisition 80 °C at 5 s. A melting curve was produced using 1 cycle at 95 °C at 30 s, 60 °C at 1 min and 95 °C at 30 s.

The variable regions V3 and V4 of the 16S rRNA gene, which are commonly used for microbial community profiling due to their high variability among bacterial taxa, was amplified with primers Bac341F (5′-CCT ACG GGN GGC WGC AG-3′) and Bac805R (5′-GAC TAC HVG GGT ATC TAA TCC-3′). The 16S rRNA gene amplification was performed according to a modified Illumina protocol. The PCR mixture contained 2 μl template DNA, 12.5 μl 2×KAPA HiFi HotStart polymerase (Kapa Biosystems, Inc., Wilmington, MA, United States), 0.5 μl 0.2 μM forward primer and 0.5 μl 0.2 μM reverse primer and 9.5 μl dH$_2$O to a final volume of 25 μl. The thermal cycling was run with an initial denaturation step at 95°C for 3 min, 30 cycles with denaturation at 95°C for 30 s, annealing at 55°C for 30 s, elongation at 72°C for 30 s and a final elongation at 72°C for 5 min. For the ITS library preparation we used primers ITS3 (5′- GATGAAGAACGYAGYRAA-3′) and ITS4 (5′-CTBTTVCCKCTTCACTCG-3′) for amplification of the ITS2 region (Toju et al., 2012). The PCR mixture contained 10 μl template DNA, 12.5 μl 2×KAPA HiFi HotStart polymerase (Kapa Biosystems, Inc., Wilmington, MA, United States), 0.5 μl 0.2 μM forward primer and 0.5 μl 0.2 μM reverse primer and 1.5 μl dH$_2$O to a final volume of 25 μl. Thermal conditions were set as follows: An initial denaturation step at 95 °C for 3 min, followed by 22 cycles with denaturation at 95 °C for 30 s, annealing at 56 °C for 30 s, elongation at 72 °C for 30 s, and a final elongation at 72 °C for 5 min. The PCR products were cleaned using 30 μl AMPure XP magnetic beads for both 16S and ITS amplicons. The second round of PCR was run with 2 μl of the amplified and cleaned PCR product for 10 cycles to incorporate overhang adapters and was run with the same conditions as the previous PCR for 16S and ITS thermal conditions, respectively. Products were cleaned, and the Nextera XT Index primers were incorporated in a third PCR reaction which was run for 8 cycles following the previous condition and an annealing temperature of 55 °C. The PCR products were quantified using a Quant-iT™ dsDNA BR assay kit on a FLUOstar

Omega fluorometric microplate reader (BMG LABTECH, Ortenberg, Germany), diluted and pooled together in equimolar ratios. The pool was quantified using the Quant-iT™ dsDNA BR assay kit on a Qubit fluorometer (Thermo Fisher Scientific, Waltham MA) and then sequenced on the Illumina MiSeq platform (Illumina, San Diego, CA) which produces two 300-bp long paired-end reads.

## 2. 7 Bioinformatic analysis

Bioinformatic analyses were performed in RStudio 4.3.0. 16S and ITS sequence reads where processed following a similar pipeline. For the 16S reads primer and adapter sequences were trimmed from the raw reads using cutadapt 0.0.1 (Martin, 2011). The important difference for the ITS dataset is the identification and removal of primers from the reads, and the verification of primer orientation and removal, due to the variable length of the ITS amplicons. This is described in the DADA2 ITS Pipeline Workflow (1.8)[1]. Forward and reverse read quality were plotted with the plotQualityProfile function from DADA2 1.21.0 (Callahan et al., 2016). Based on the read quality a trimming of 280 bp and 200 bp were set for the forward and reverse reads, respectively, (for the 16S reads) using FilterAndTrim, according to their quality (Callahan et al., 2016) whereas no trimming was done for the ITS sequences due to their variable length. Error models were built for the forward and reverse reads, followed by dereplication and clustering into amplicon sequence variants (ASVs) with DADA2 (Callahan et al., 2017). The denoised forward and reverse reads were merged using the function mergePairs with default parameters with a minimum overlap of 12 nucleotides, allowing 0 mismatches. Sequence tables were made with the function makeSequenceTable. ASVs shorter than 400 and longer than 430 nucleotides were removed from the 16S whereas all sequences were kept for the ITS dataset. Chimeric sequence removal was done using the removeBimeraDenovo function and taxonomic assignment was accomplished using the naive Bayesian classifier against the SILVA ribosomal RNA gene database v138 (Quast et al., 2012) for the 16S rRNA sequences while the UNITE database was used to classify the ITS sequences (Koljalg et al., 2005). ASVs mapped to mitochondria and chloroplasts were removed from the 16S dataset. Samples were decontaminated using the prevalence method (Threshold = 0.1) from the Decontam package (Davis et al., 2018) using the DNA extraction negative control and PCR Negative control as the control group. Statistical tests and visualization of the data was performed with phyloseq (Mcmurdie and Holmes, 2013), Vegan (Dixon, 2003) and microeco (Liu et al., 2021).

## 3 Results and discussion

### 3.1 Highly active INPs in Northeast Greenlandic soils

By quantifying INP concentrations in soil samples (Fig. 2), we found that the onset of freezing was high (>-8°C) at all locations (Supplementary Fig. 1). The ice nucleation site density per gram of soil particles <63 μm ($N_m$) as a function of temperature is

---

[1] https://benjjneb.github.io/dada2/ITS_workflow.html

depicted in Fig. 2 for all locations. INP-10 values varied by 2 orders of magnitude between locations, from $3.19 \cdot 10^4$ g$^{-1}$ soil (West 4) up to $1.55 \cdot 10^6$ g$^{-1}$ soil (West 1), which is also reflected in the INP-15 values (Table 1).

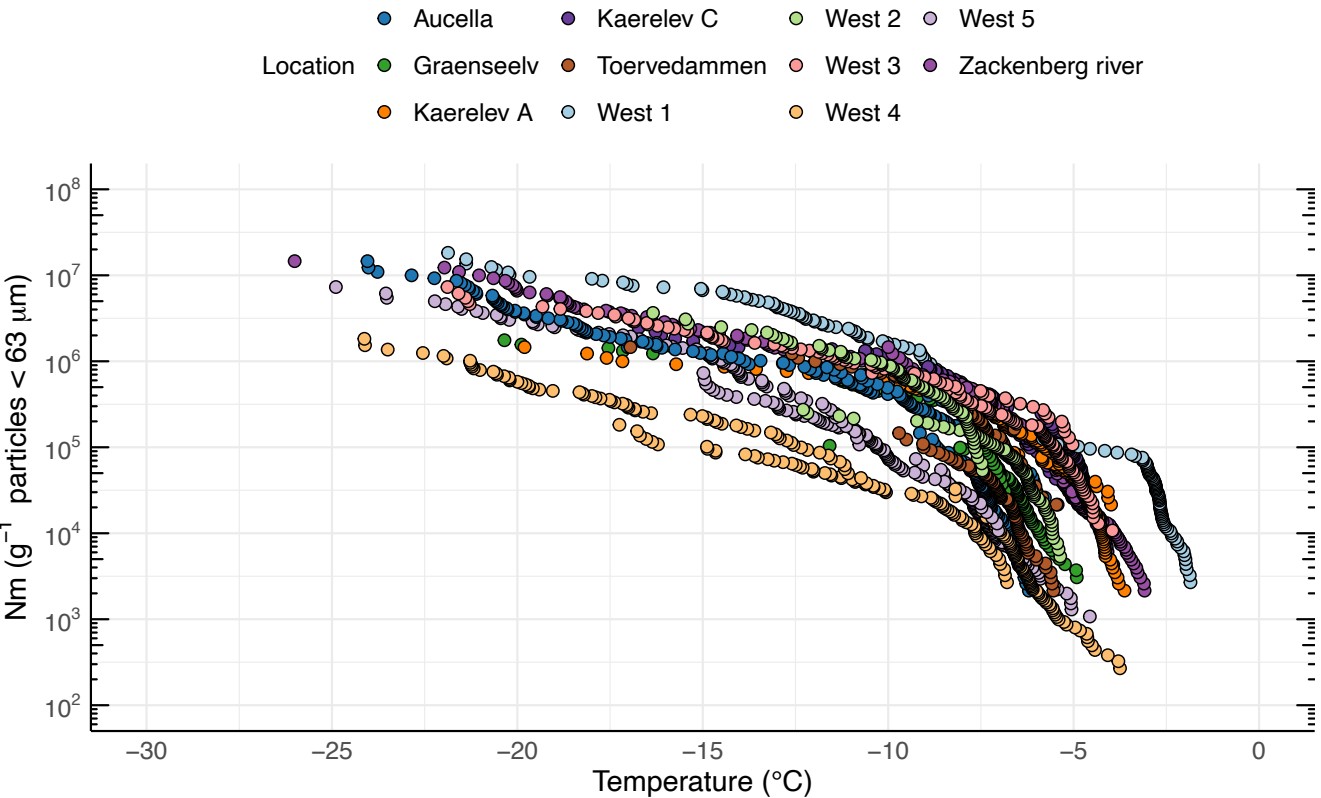

**Figure 2** INP concentrations (i.e. ice nucleation site density per gram of soil <63 μm ($N_m$)) as a function of temperature in soil samples from eleven different locations in Northeast Greenland.

Previous investigations showed that INPs in soils exhibited activity at temperatures above -15 °C and contain biological residues, both in temperate and Arctic regions (Conen et al., 2011; Huang et al., 2021). Several studies reported that INPs are more enriched in soils of colder climates compared to warmer climates (Creamean et al., 2020; Schnell and Vali, 1976; Tobo et al., 2019). A comparison of INP concentration with previously published studies in the Arctic, is shown in Fig. 3. Previous studies focused on similar latitudinal geographical regions while differing in longitude, ranging from the Utqiagvik (USA) to Novosibirsk (Russia), Central Yakutia (Russia) and Svalbard (Norway). Soil INP-10 concentrations ranged from $10^8$-$10^9$ g$^{-1}$ soil in Russia (Conen et al 2011, 2018) ~$10^8$ g$^{-1}$ soil in glacially outwashed sediments in Svalbard (Tobo et al. 2019) and from >$10^8$ g$^{-1}$ down to $10^6$ g$^{-1}$ in permafrost soils from Utqiagvik (Barry et al 2023). In this study, we found the lowest INP-10 concentrations (ranging from ~$10^4$ g$^{-1}$ to ~$10^6$ g$^{-1}$). A possible explanation for the lower concentration could be the rather low

carbon content of the soils measured in this study with 9 out of 11 samples <5 % w/w TC, which is less than what Conen and Yakutin (2018) found in soils from Central Yakutia, but similar to the glacial outwash sediment that Tobo et al. (2019) investigated (Conen and Yakutin, 2018; Tobo et al., 2019). Higher carbon content in soils usually reflects higher amount of biomass and microbial diversity present in the soil (Bastida et al., 2021). We also found significantly more biomass (16S rRNA copies) in soils with higher soil carbon content (R =0.8, p = 0.0052). When relating the TC with the $INP_{-10}$ concentrations in the soil, we found a non-significant negative correlation (R = -0.33, p = 0.327). Although, carbon content has been proposed as an explanatory variable for $INP_{-10}$ concentrations, reflecting higher microbial productivity, studies have found no simple correlation between the two (Conen et al., 2011; Tobo et al., 2019). The higher INP concentrations reported by Barry et al. (2023) may partly stem from their methodology, where active layer soil was prepared by thawing and stirring without drying and sieving, directly suspending the bulk soil in water. This likely resulted in a broader representation of the particle size spectrum, compared to our approach, where we used freeze-drying, and comminuting in a mortar before differential settling focusing on isolating particles smaller than 63 $\mu$m. Similarly, both Tobo et al. (2019) and Conen et al. (2011, 2018) used air-drying to prepare soils before sieving through 90 $\mu$m and 63 $\mu$m meshes, respectively, to isolate finer fractions. They then further separated particles smaller than 5 $\mu$m using differential settling combined with filtration. The large variations in $INP_{-10}$ between our study and results obtained by Tobo et al (2019) and Conen et al. (2011) may be due by differences in microbial community composition and activity or in washout rates, which can be influenced by factors such as soil porosity, and permeability (Wen et al., 2022). Our results showed higher onset temperatures (between -1.5 °C and -4.7 °C) compared to previous studies of Arctic soils (Fig. 3). Methodological differences, such as droplet volume used in freezing assays, must be considered when interpreting this trend. Studies using smaller volumes (e.g. 5 µL in Tobo et al., 2019), have a lower sensitivity and cannot be directly compared to our study. However, several studies used larger droplet volumes (e.g., 50 µL in Conen et al., 2018; Barry et al., 2023b, and 100 µL in Conen et al., 2012), which have a higher sensitivity than the micro-Pinguin assay and a comparable potential to detect rare highly active INPs. Therefore, the higher freezing onset that we observe does not seem to only be linked to methodological differences but reflects differences in the INP populations in these environments. INPs active at such high temperatures are generally proteinaceous (Santl-Temkiv et al., 2022; Kanji et al., 2017; Huang et al., 2021) and are often associated with microbial sources, including bacteria and fungi (Barry et al., 2023b; Tobo et al., 2019; Conen et al., 2011). The presence of higher onset temperatures in this study compared to other studies may indicate differences in either the identity or in the activity of their microbial producers across Arctic terrestrial environments. The compiled results indicate the presence of highly active INPs in Arctic soils, raising the potential for hydrological transport into river systems or oceans, as proposed by (Creamean et al., 2020).

**Table 1** Sampling location details, $T_{50}$ values for streams and soils, and $INP_{-15}$ and $INP_{-10}$ in streams and soils respectively for the twelve locations in Northeast Greenland.

| | | | Streams | | | Soil | | |
|---|---|---|---|---|---|---|---|---|
| Location | Date | Latitude & Longitude | $T_{50}$ (°C) | $INP_{-10}$ (mL$^{-1}$) | $INP_{-15}$ (mL$^{-1}$) | $T_{50}$ (°C) | $INP_{-10}$ (g$^{-1}$) | $INP_{-15}$ (g$^{-1}$) |
| Kærelv C | 08-09-21 | 74.47102° N -20.51882° W | -8.08 | $4.12 \cdot 10^2$ | $6.58 \cdot 10^3$ | -4.77 | $1.07 \cdot 10^6$ | $1.98 \cdot 10^6$ |
| Kærelv A | 08-09-21 | 74.49468° N -20.49629° W | -7.11 | $1.10 \cdot 10^3$ | $9.59 \cdot 10^3$ | -4.58 | $5.57 \cdot 10^5$ | $9.91 \cdot 10^5$ |
| West 5 | 10-09-21 | 74.45419° N -20.85248° W | -7.88 | $1.08 \cdot 10^3$ | $1.51 \cdot 10^4$ | -7.27 | $7.88 \cdot 10^4$ | $1.02 \cdot 10^6$ |
| West 4 | 10-09-21 | 74.44862° N -20.78540° W | -9.72 | $2.44 \cdot 10^1$ | $1.57 \cdot 10^4$ | -6.21 | $3.19 \cdot 10^4$ | $1.64 \cdot 10^5$ |
| Aucella | 12-09-21 | 74.50327° N -20.54179° W | -6.76 | $8.66 \cdot 10^2$ | $6.01 \cdot 10^3$ | -7.28 | $4.42 \cdot 10^5$ | $1.21 \cdot 10^6$ |
| Tørvedammen | 12-09-21 | 74.47926° N -20.55715° W | -7.90 | $2.90 \cdot 10^3$ | $3.88 \cdot 10^4$ | -6.77 | $5.91 \cdot 10^5$ | $1.37 \cdot 10^6$ |
| West 1 | 12-09-21 | 74.47394° N -20.60757° W | -7.39 | $4.88 \cdot 10^3$ | $4.15 \cdot 10^5$ | -2.72 | $1.55 \cdot 10^6$ | $5.94 \cdot 10^6$ |
| West 3 | 13-09-21 | 74.46048° N -20.69274° W | -8.39 | $3.95 \cdot 10^1$ | $1.96 \cdot 10^3$ | -5.66 | $7.81 \cdot 10^5$ | $2.17 \cdot 10^6$ |
| West 2 | 13-09-21 | 74.47158° N -20.65557° W | -9.52 | $4.38 \cdot 10^1$ | $1.29 \cdot 10^3$ | -7.11 | $7.47 \cdot 10^5$ | $2.76 \cdot 10^6$ |
| Grænseelv | 15-09-21 | 74.47321° N -20.50832° W | -9.09 | $7.16 \cdot 10^1$ | $1.08 \cdot 10^4$ | -6.57 | $4.77 \cdot 10^5$ | $1.13 \cdot 10^6$ |
| Zackenberg river | 17-09-21 | 74.46992° N - 20.57417° W | -7.95 | $5.88 \cdot 10^2$ | $1.48 \cdot 10^4$ | -4.89 | $9.23 \cdot 10^5$ | $2.13 \cdot 10^6$ |
| Jurassic 1 | 19-09-21 | 74.43010° N -20.30258° W | -9.49 | $5.42 \cdot 10^1$ | $2.30 \cdot 10^3$ | - | - | - |

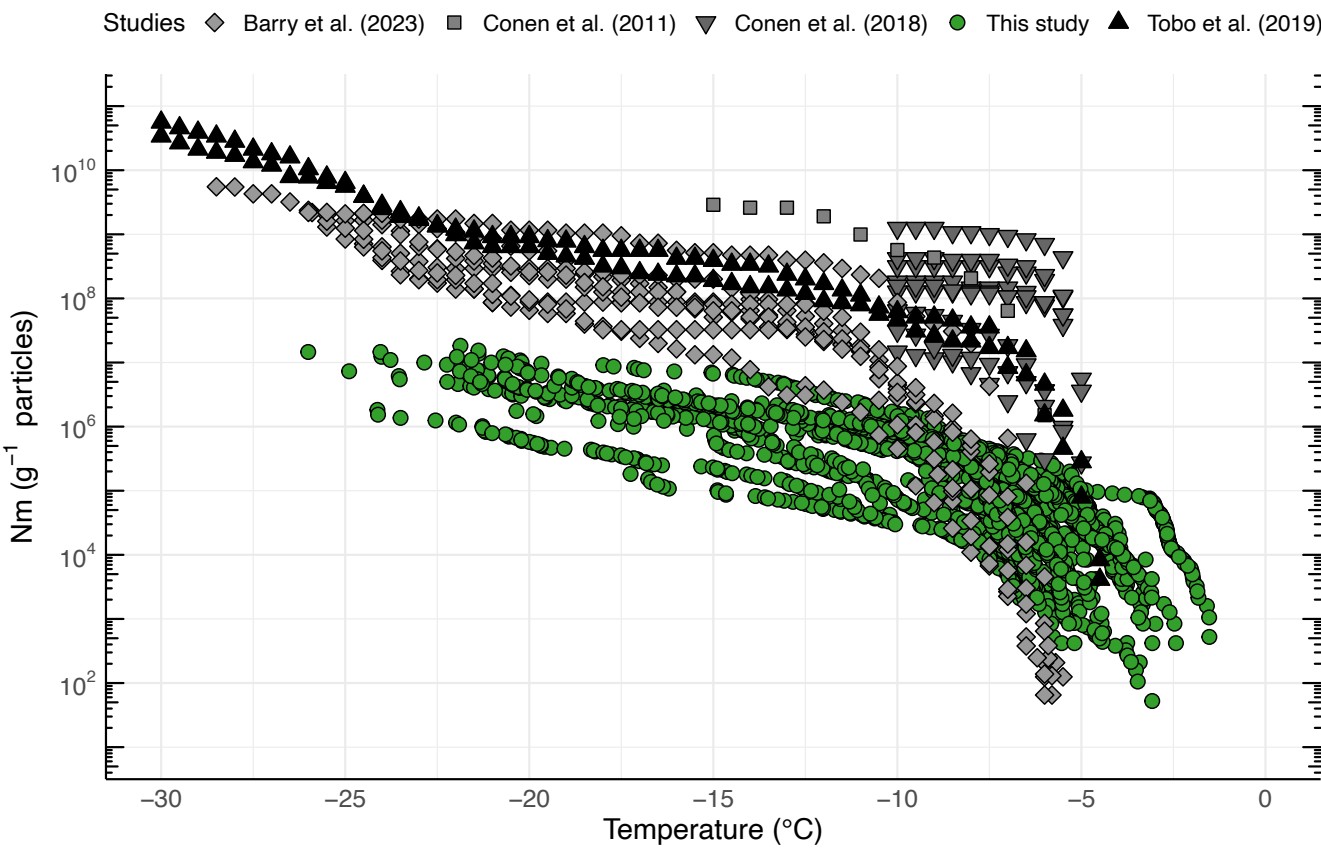

**Figure 3** Ice nucleation site density per gram of soil ($N_m$) as a function of temperature from 5 different studies. Light grey diamonds derived from Barry et al. 2023 (Utqiagvik - USA), grey squares from Conen et al. 2011(Novosibirsk – Russia), dark grey inverted triangles from Conen et al. 2018 (Central Yakutia – Russia), black triangles from Tobo et al. 2019 (Svalbard – Norway), and green circles from this study (Zackenberg – Greenland).

### 3.2 BioINP in northeastern Greenlandic soils are diverse and potentially originate from different sources

To further characterize INPs within the Arctic soil, we used filtration analysis as different microorganisms produce INpro of different molecular sizes, which can either be firmly bound to the cells or easily removed resulting in soluble proteins (O'sullivan et al., 2015; Santl-Temkiv et al., 2022). A similar approach has previously been used to study the origin of INP in environmental samples (Conen and Yakutin, 2018; Fröhlich-Nowoisky et al., 2015). A Kruskal-Wallis test, indicated significant differences between the filtration treatments (p-value = 0.0001) (Fig. 4). A Wilcoxon rank sum test showed a

significant difference between the bulk sample and the 300-100 kDa fraction (p = 0.0046). Subsequently, we analyzed the samples from individual locations to identify specific patterns.

.

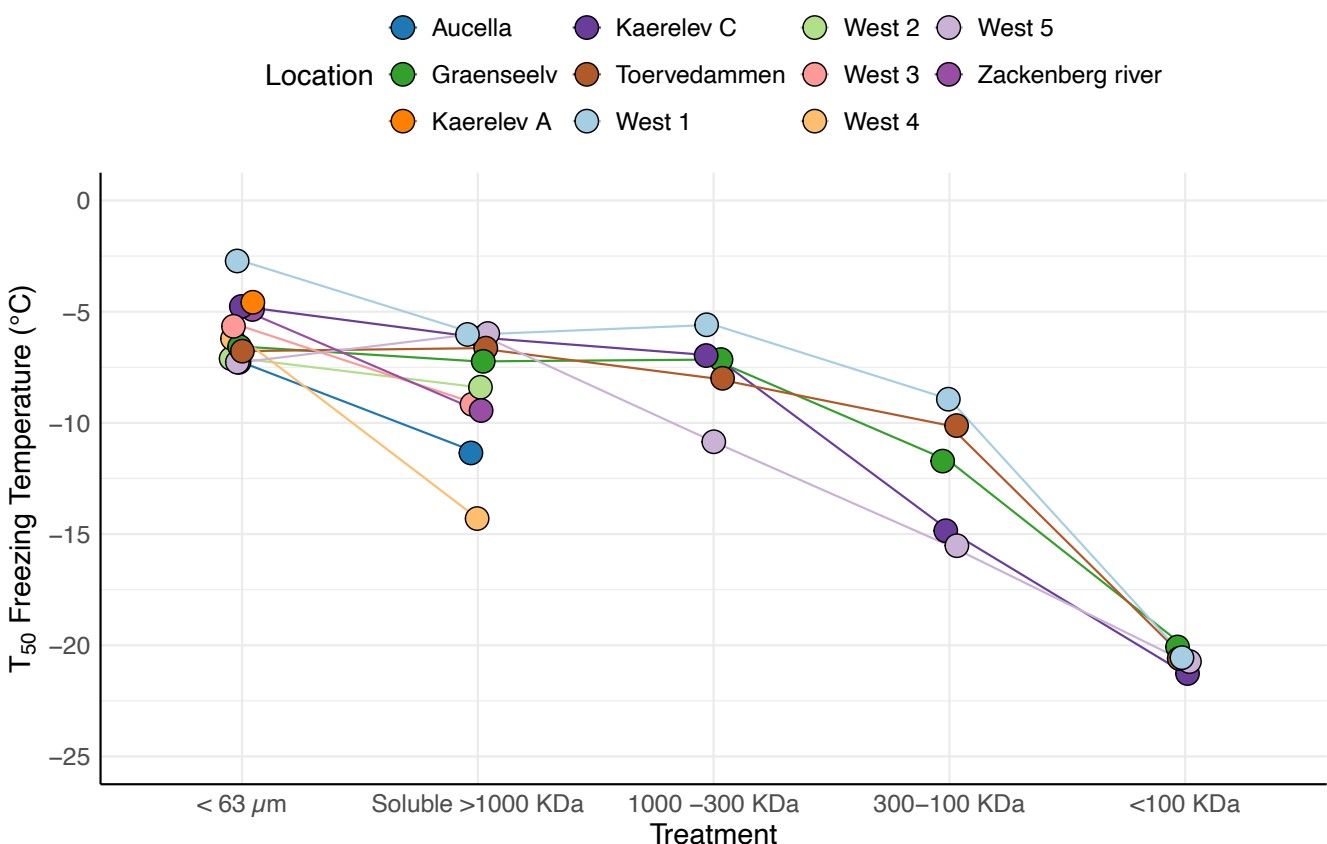

**Figure 4** Freezing activity ($T_{50}$) for soil samples collected from eleven different locations in Northeast Greenland. The samples underwent various filtration treatments. The "< 63 μm" category represents the untreated sample. The "Soluble > 1000 kDa" category indicates samples that passed through a 0.2 μm filter but were retained on a 1000 kDa filter. The "1000-300 kDa" category comprises samples that passed through the 1000 kDa filter but were retained on the 300 kDa filter. The "300-100 kDa" category includes samples that passed through the 300 kDa filter but were retained on the 100 kDa filter. Finally, the "<100 kDa" category encompasses samples that passed through the 100 kDa filter. Note that the samples from Aucella, Zackenberg and West 2, 3 and 4 only were passed through the 0.2 μm filter as they already lost most of their high activity after this treatment.

The analysis reveals significant variations in the inferred composition of INPs within the soil across different locations (Fig. 4). There was a clear decrease in $T_{50}$ activity following the filtration of samples through a 0.2 μm filter in certain locations, such as West 4 and Aucella, which could be explained by a predominant association of INPs with microbial cells, or with soil

particles. O'Sullivan et al. (2016) demonstrated that INPs bind to clay particles smaller than 11 μm. They additionally found that typical topsoil pH levels (between 7.4 and 4.6) do not significantly affect this absorption. However, the presence of electrolytes such as $CaCl_2$ and $MgCl_2$, along with the contact time between clay particles and INPs, enhances absorption. Additionally, these clay particles can be aerosolized directly from the soil (O'sullivan et al., 2016). Our study found that soil particle size distribution was relatively homogeneous across different locations, with approximately 15-20% of soil particles

being clay particles smaller than 5 μm (Supplementary Fig. 2). This suggests that a portion of INPs would bind to these clay particles, allowing them to be transported directly into the atmosphere or washed into streams. Alternatively, the loss of activity may be explained by the presence of bacterial membrane-bound INpros (Santl-Temkiv et al., 2022). Conversely, in samples from most locations soluble INP were >1000 kDa, which is in agreement with the formation of large bacterial INpro aggregates/oligomers within the 700 to 2000 kDa range (Schmid et al., 1997). In the sample from Kaerelev C, we observed

predominant INPs in the size range between 300 and 1000 kDa, which is consistent with INpro produced by *Fusarium acuminatum*. The size of INpro molecules from *F. acuminatum* have been demonstrated at ~5.3kDa INPs with aggregates reaching ~700 kDa (Schwidetzky et al., 2023a). Finally, samples from Toervedammen, West 1, and Graenseelv, contained predominant INP with sizes between 100 and 300 kDa filter. Previous studies by Fröhlich-Nowoisky et al. (2015) and Kunert et al (2019) suggest that *Mortierella sp. and Fusarium sp.* produce a INpro which are between 300-100 kDa (Kunert et al.,

2019). The gradual loss of INPs during filtration at the different locations suggests a mixture of different-sized INPs, predominantly originating from fungi and bacteria (Fig. 4). This interpretation is supported by the fact that fungal INPs are known to span a wide size range, including small <100 kDa (e.g., 5 kDa), and medium-sized molecules (100-300 kDa and 300-1000 kDa), and can bind to clay particles, resulting in INPs >1000 kDa and >0.2 μm (Kunert et al., 2019; Schwidetzky et al., 2023b; O'sullivan et al., 2015; Conen and Yakutin, 2018). The size distribution, combined with the observed solubility and

INA, aligns with the characteristics of fungal INPs, further supporting their major contribution to the INP pool in these soils. Our findings somewhat align with Barry et al. (2023b), who demonstrated that the majority of INPs in soil were larger than 0.2 μm and primarily of biological origin. Furthermore, their heat treatment experiments revealed that INPs in permafrost soil are predominantly heat-labile, further supporting their biological nature. Interestingly, Barry et al. (2023b) reported a limited presence of soluble INPs smaller than 0.2 μm in permafrost soil, suggesting a scarcity of such low-molecular-weight biological

INPs in their study system. This contrasts with the observed presence of INPs spanning a large range of sizes in our samples, including <100 kDa and aggregates >1000 kDa (Fig. 4). Such discrepancies might reflect differences in the environmental conditions, microbial communities, or soil composition between their study sites and ours. Additionally, Barry et al. (2023a) highlighted that INP concentrations in permafrost soil are influenced by particle size and composition, observing that larger particles (>10 μm) were significant contributors to INA in younger permafrost samples. While their findings pertain to larger

particle fractions, our study emphasizes the role of smaller clay-bound particles (<5 μm) for transporting INPs into the

atmosphere. While larger particles settle quickly after aerosolization, smaller, clay-bound INPs are more prone to longer atmospheric residence time after aerosolization (Meinander et al., 2022).

The size distribution, combined with the observed solubility and INA, aligns with the characteristics of fungal INPs, further supporting their contribution to the INP pool in these soils. The INPs identified in Arctic soils have potential implications for atmospheric processes. Given that soil particles, including those bound with INPs, can become aerosolized through wind or other disturbances, these particles might contribute to atmospheric INP concentrations. The binding of INPs to clay particles in soils suggests that these particles could be transported to the atmosphere. Previous research has shown that soil dust and soil microorganisms can become a significant source of atmospheric aerosols in both temperate and the Arctic region (O'sullivan et al., 2016; Tobo et al., 2019; Santl-Temkiv et al., 2018). Consequently, the high concentration of INPs in soil may enhance the INA in the atmosphere, potentially impacting cloud formation and regional climate. Our findings align with studies indicating that INPs in soil environments can contribute to atmospheric INP levels. For instance, increased INP concentrations in Arctic soils might be linked to higher upward fluxes of atmospheric INP levels above these soils, similar to observations in other environments, e.g., arid soils, deserts and agricultural soil (Kanji et al., 2017). This suggests that understanding soil INP sources is crucial for assessing their potential impact on atmospheric processes.

### 3.3 The link between the microbial community members and INPs

To link microbial community members to the INPs present in soils, we analyzed the bacterial and fungal communities in the soil samples. We found between $1.56 \cdot 10^7$ and $7.82 \cdot 10^9$ 16S rRNA gene copies $\cdot$ g$^{-1}$ of soil, which is within the same order of magnitude as found in similar soils (Ganzert et al., 2014; Santl-Temkiv et al., 2018). In total, we found that the bacterial community consisted of 6579 unique amplicon sequence variants (ASVs) (320-438 ASVs), which is on the lower side of the range found in soil bacterial communities across the Arctic region (Malard et al., 2019). The ASVs were dominated by members belonging to the phyla Proteobacteria ($27.22 \pm 4.86$ %) and Actinobacteria ($18.60 \pm 9.46$ %), while Planctomycetota, Bacteroiota, Acidobacteriota and Verrucomicrobiota each accounted for ($\sim 8 \pm 4.66\%$) of the total community. Also, members of the following phyla were present in the samples: Cyanobacteria ($5.3 \pm 7.12\%$), Chloroflexi ($5.29 \pm 5\%$), Myxococcota ($3.03 \pm 1.92\%$), and Gemmatimonadota ($2.32 \pm 1.08\%$) (Supplementary Fig. 3). Our results agree with community data reported in Malard et al (2019) showing consistency of the overall bacterial composition in Arctic soils. By determining the sequences of the ITS region, we identified 2376 unique ASVs (63-234 ASVs) in total, which is comparable to the composition of fungal communities previously reported in Arctic soils (Varsadiya et al., 2021; Dziurzynski et al., 2023). The phylum Ascomycota was the most abundant phylum in all analyzed samples. They accounted for 60-70 % of all ASVs in 7 out of 10 locations (Supplementary Fig. 4). A considerable fraction ($23.6 \pm 19.8\%$) of the ASVs could not be affiliated to a specific phylum. The additional phyla were Basidiomycota ($12.9 \pm 12.4\%$), Chytridiomycota ($2.4 \pm 2.6\%$), and Mortierellomycota ($1.2 \pm 2.4\%$). We did not find any distinct geographical pattern for the composition of bacterial or fungal

communities (Fig. 1 and supplementary Figs. 3 and 4). Searching for known producers of INPs in the different soil samples e.g., *Pseudomonas, Erwinia, Pantoea, Xanthomonas*, and *Lysinibacillu*s, we only found 16S rRNA sequences related to *Pseudomonas*. These sequences were present at three locations at very low proportions (<0.0006%). Using a similar approach for fungi, we searched the presence of ITS region sequences affiliated with *Fusarium, Mortierella, Acremonium, Isaria, and Puccinia*, which were previously reported to produce INPs (Pouleur et al., 1992; Fröhlich-Nowoisky et al., 2015; Huffman et al., 2013; Morris et al., 2013). Only sequences affiliated to the genera, *Acremonium* and *Mortierella* were present in our dataset. *Acremonium* was only detected at one location, while *Mortierella* was present at most locations, albeit at a relatively low abundance (<0.0025%) (Supplementary Fig. 5). Recent research has shown that INpro remain stable in permafrost soil for over 30,000 years and can retain their activity after thawing (K. R. Barry et al., 2023; Creamean et al., 2020). Thus, INpro could had also been produced by past microbial communities and have accumulated in soils over time. Both habitat properties and plant cover were shown to affect abundance and diversity of *Mortierella* sp. in soils (Telagathoti et al., 2021; Mannisto et al., 2024; Shi et al., 2015). Arctic soils were typically found to contain a much higher fraction (on the order of 1-10%) of *Mortierella* sp. compared to our study (Mannisto et al., 2024; Varsadiya et al., 2021) As Arctic terrestrial environments are undergoing dramatic changes including changes in snow cover, soil development, and associated vegetation responses, e.g. the Arctic greening, soil microbial community composition is affected (Doetterl et al., 2021). Thus, past fungal communities at the sites we describe may have featured a higher abundance of *Mortierella,* which have produced the observed INpro over time. Hence, as sequences affiliated to *Mortierella* were found at most locations, INA *Mortierella* sp. may had produced the highly potent INpro present in soils. This conclusion fits with filtration experiments that showed presence of INpro in the different soluble fractions at several locations, consistent with what has been shown for fungal INpro. While particulate INpro present at other locations could be associated with bacterial INpro, this is not supported by the fact that bacterial genera known to produce INpro were not present consistently, and that bacterial INpro have high temporal turnover rates (Watanabe et al, 1990). Thus, the particulate INpro may be more likely associated to fungal INpro bound to clay particles.

An alternative explanation is that the INPs in the soils were produced by microorganisms not yet identified in the literature to produce bioINP. Over the past years it has become evident that the production of biogenic INA material is widespread through the tree of life, encompassing bacteria, algae, fungi, lichens, and pollen (Eufemio et al., 2023; Santl-Temkiv et al., 2022; Tesson and Santl-Temkiv, 2018; Adams et al., 2021; Gute et al., 2020). Based on these findings, a heterogenous substrate like soil could potentially host many different taxa capable of producing INA material. Consequently, we used a bulk approach to link the high ice-nucleation activity in the samples and the bacteria and fungi present in the soil. Using Spearman's rank correlation, we found that 14 bacterial and 3 fungal genera significantly positively correlated with the INP$_{-10}$ and INP$_{-15}$ concentration (Fig. 5). Interestingly, none of the taxa correlating with INP$_{-10}$ and INP$_{-15}$ were related to known INP producers. These taxa could be previously unknown INpro producers. Their isolation from soil would be needed to establish whether they have the ability to produce INpro and equivocally support the observed correlation. The bacteria were diverse and included phyla like, Actinomycetota, Pseudomonadota, Verrucomicrobiota, Bacteroidota, Planctomycetota as well as Cyanobacteria highlighting the remarkable diversity of potential INP producers in soil ecosystems. While the Spearman rank correlations

provided insights into potential associations between specific microbial genera and the concentration of INPs in soil samples, it is essential to acknowledge the limitations of this statistical approach. Correlations do not imply causation, and the observed relationships may be influenced by various confounding factors. Additionally, the Spearman rank correlation assesses monotonic relationships but may not capture nonlinear associations.

Fungal sequences that positively correlated with INP$_{-10}$ and INP$_{-15}$ concentrations affiliated with the phylum Ascomycota. While this phylum is known to encompass many different lifestyles, only saprotrophic, pathogenic and lichenized fungi are known to produce INPs (Pouleur et al., 1992; Fröhlich-Nowoisky et al., 2015; Huffman et al., 2013; Morris et al., 2013). To understand the trophic mode and guilds of the potential fungal INP-producers that we identified, we assigned these using the FungGuild database (Nguyen et al., 2016). We discovered that the predominant groups included fungi with different functions, comprising ectomycorrhizal fungi, wood saprotrophs, plant pathogens and lichenized fungi (Supplementary Fig. 6). This is in good agreement with Varsadiya et al. (2021) who also found that the ectomycorrhizal lifestyle dominated in Arctic top soil (Varsadiya et al., 2021). The fungal genus *Atla*, which shows a significant positive corrrelation with INP$_{-10}$ and INP$_{-15}$ number concentrations (see Fig. 5), was initially identified as a lichen by Savic and Tibell (Savić and Tibell, 2008). Lichens, known for their high ice nucleation activity, may contribute to the production of INPs in Arctic soil ecosystems. However, our knowledge on the structure of INPs produced by lichens are limited. Recent studies have shown that they are active at high temperatures (-3 °C), that they are not of bacterial origin (Moffett et al., 2015) and that they differ in their heat stability at 98 °C with class A denaturation while class C retaining activity which could suggest a combination of proteinaceous and polysaccharide based INPs (Eufemio et al., 2023).

The link between microbial communities and INPs has significant implications for atmospheric INP sources. Specific bacterial and fungal genera residing in soils are known to produce high-temperature proteinaceous INPs, making soil a significant source for atmospheric INPs emitted through wind erosion (Cornwell et al., 2023; Santl-Temkiv et al., 2022) and thus contributing to atmospheric INP concentrations (O'sullivan et al., 2016; Bullard et al., 2016). Understanding the specific microbial taxa and their metabolic activities responsible for INP production in soils and streams is crucial for predicting their contribution to atmospheric ice nucleation processes. This knowledge enables us to assess their impact on cloud formation, precipitation patterns, and regional climate dynamics (Keuschnig et al., 2023). Microbial community composition, including the abundance and activity of microbial strains that produce proteinaceous INPs, significantly changes as a response to environmental conditions, such as pH, organic matter content, nutrient availability, water activity and Arctic greening (Malard et al., 2022; Wong et al., 2023). Thus, identifying key microbial INP producers and understanding their ecology and activity may have major implications for predicting the potential of different Arctic soils as reservoirs for INPs. This knowledge will feed into the potential to predict atmospheric INP concentrations and their overall impact on atmospheric processes and climate change.

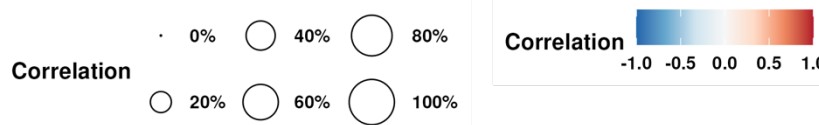

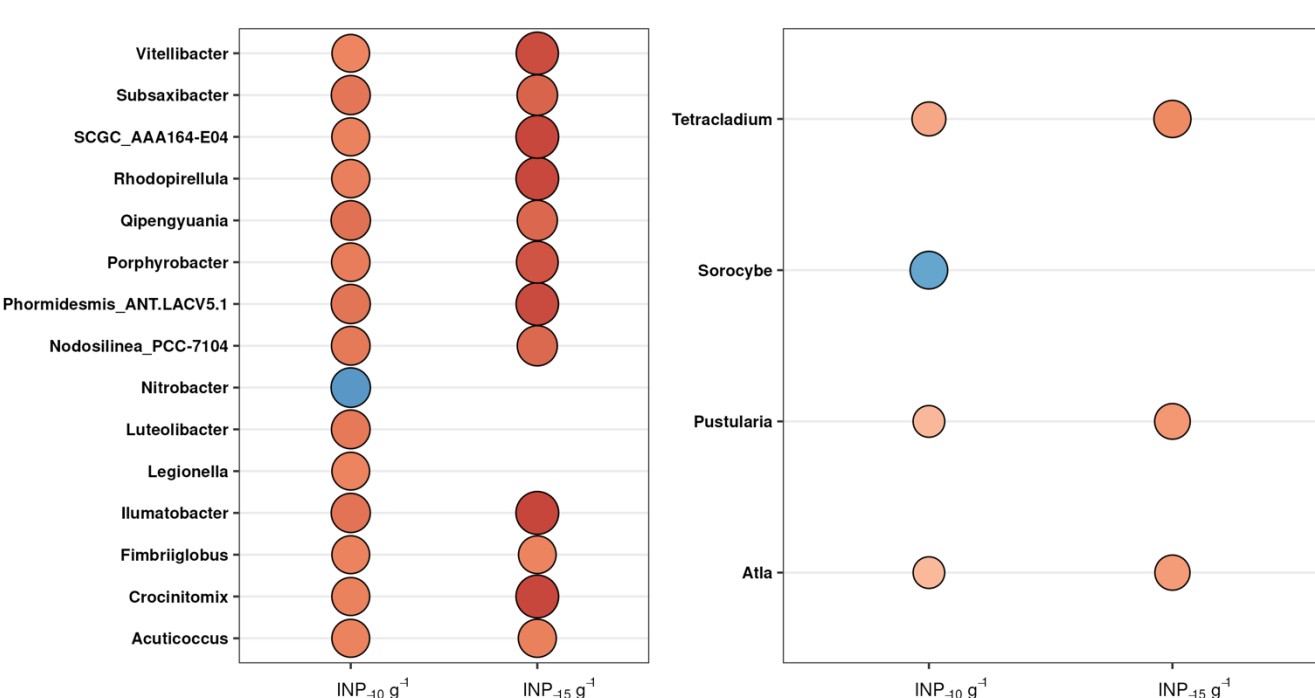

**Figure 5** Spearman's rank correlations between the concentrations of INP$_{-10}$ g$^{-1}$ and INP$_{-15}$ g$^{-1}$ smaller than 63 $\mu$m and the relative abundance of genera of bacteria (left) and fungi (right). The size of the bubbles represents the strength of the correlation
coefficient, while the color indicates whether the correlation is positive (red) or negative (blue). We have excluded the correlations with INP15 that are significant but have no significant correlation with INP10, focusing only on the most intriguing taxa. To account for false positives and minimize false negatives, all P-values have been adjusted using the Benjamin and Hochberg method (FDR).

### 3.4 Characterization of INPs in Greenlandic streams

In addition to characterizing soil INPs and their potential sources, we investigated the linkages between soil-freshwater INPs. The freezing onset was > -10°C for all twelve water sampling locations (Supplementary Fig. 7). The highest onset temperature was found in Aucella (-5.9°C) and lowest in West 4 (-9.1°C). The high freezing temperatures indicate that the INPs are of biological origin (Kanji et al., 2017). The ice nucleation site density per volume of freshwater (N$_V$) as a function of temperature is shown in Fig. 6. The INP$_{-10}$ concentration measured in our study (average: 1005 mL$^{-1}$; range 24-4,880 mL$^{-1}$) (Table 1) are
much lower than those reported by Barry et al. (2023a) for Arctic thermokarst lakes (average: 34300 mL$^{-1}$; range 1360-242,000

mL$^{-1}$). One potential explanation for this difference is the higher soil INP concentrations reported in Barry et al.'s study compared to our measured soil INP concentrations. If soil is a major source of INPs to freshwater systems, as suggested by both studies, then lower soil INP concentrations in our sampling locations may directly contribute to the lower INP concentrations observed in Arctic streams relative to thermokarst lakes.

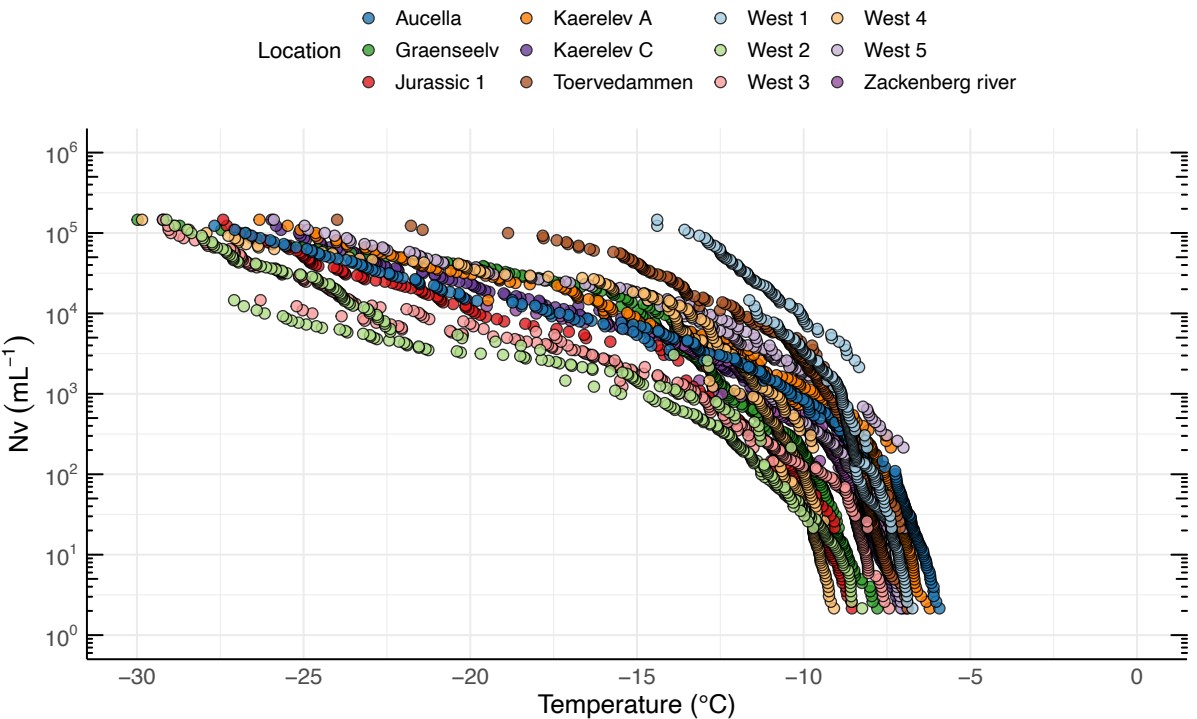

**Figure 6** Freezing activity and INP concentrations for freshwater samples from twelve different locations in Northeast Greenland showing the ice nucleation site density per volume (*Nv*) as a function of temperature.

Our results agree with what has been reported in previous studies suggesting that INP$_{-10}$ are omnipresent in freshwater systems (Larsen et al., 2017; Knackstedt et al., 2018; Moffett, 2016; Moffett et al., 2018). Since no Arctic rivers have previously been
investigated, we have turned towards comparisons with temperate regions which showed similar onset freezing temperatures and INP$_{-10}$ concentration as we report here. Thus, a survey of a major river system in the USA (the Mississippi, Missouri, Platte, and Sweetwater Rivers) found that the freezing onset varied between -4°C and -6°C, and the INP$_{-10}$ number concentration ranged from 87 to 47,000 mL$^{-1}$ (Moffett et al., 2018). A study of the River Gwaun in Wales found freezing onsets between -3°C and -5°C, and INP$_{-10}$ number concentration that ranged from 539 to 1570 mL$^{-1}$ (Moffett, 2016).
Additionally, INP$_{-10}$ in the Maumee River (Ohio, the USA) exceeded 10,000 mL$^{-1}$ (Knackstedt et al., 2018), while INP$_{-10}$ in the Rhine River (Switzerland) occasionally exceeded 1000 mL$^{-1}$ (Larsen et al., 2017). The high concentrations of INP$_{-10}$ in Arctic streams align with findings that several Arctic environments are rich in INPs, emphasizing their potential contribution

as a regional source. However, despite these abundant local sources, aerosol INP concentrations in the Arctic atmosphere remain relatively low, possibly due to transport and deposition dynamics (Huang et al., 2021). The concentrations of INPs in stream water are greater than typically found in marine systems. The average INP$_{-10}$ found here was at least seven times greater than previously reported in Arctic marine systems, where INP$_{-10}$ have been found to be between 10 and 100 mL$^{-1}$ (Irish et al., 2017). A recent study has estimated that the total freshwater discharge in Greenland has increased from 136 Gt · yr$^{-1}$ in 1992 to 785 Gt · yr$^{-1}$ in 2012 (Mankoff et al., 2020). This suggests that freshwater is an increasing source of INPs in the Arctic oceans and that streams serve as a transport mechanism from soil into the ocean, or directly into the atmosphere from turbulence in the streams, with increased thawing due to Arctic amplification (Rantanen et al., 2022).

To further characterize stream INPs, we used filtration experiments to compare with results obtained for the adjacent soil sampling sites. For most samples, T$_{50}$ was higher for the filtered samples than for the non-filtered samples (Fig. 7). The same phenomenon has previously been described by Baloh et al. (2021) in river and pond samples in Obergurgl, Austria. They argue that a reasonable explanation is that some material in the samples counteract ice nucleation, e.g., antifreeze proteins, and that these are removed by filtration thereby increasing the ice nucleation activity in the filtrate (Baloh et al., 2021). This seems unlikely, since antifreeze proteins are typically small and soluble, hence they would pass through the 0.22 μm filter (Davies, 2014; Lorv et al., 2014). Another possible explanation is that filtration through a 0.22 μm filter causes cell lysis. This could either release cell content or break membranes apart carrying INPs, thus the number of INPs would be increased in the filtrate. Bacterial and fungal cells can release INPs (Fröhlich-Nowoisky et al., 2015; Pummer et al., 2015; Phelps et al., 1986), and some INPs are associated with phytoplanktonic exudates (Wilson et al., 2015). The results from the filtration experiments are comparable to other studies suggesting that a majority of INPs in rivers and streams are soluble (Larsen et al., 2017; Moffett et al., 2018; Knackstedt et al., 2018). Statistical analysis using the Kruskal-Wallis test showed a significant difference among the treatments (p-value = $1.721 \cdot 10^{-8}$). Post hoc Wilcoxon rank sum tests revealed a significant change from bulk to the 300-100 kDa category (p = 0.0021). This was also observed in soil samples (Fig. 4), indicating that similar INPs are present in soil and streams which further imply the possible transfer of INPs from soil into the streams.

 "Aucella" and "Jurassic 1" streams showed the presence of INP >1000 kDa, indicating the presence of bacterial INpro or fungal INpro bound to soil particles. Conversely, other streams contained mostly INP <1000 kDa. Streams like "Kaerelev C" contained mostly INPs within the 300-1000 kDa range and streams like "Graenselev" INPs between 100 and 300 kDa, both consistent with fungal-like INpros seen in soil samples. In conclusion, INPs in the streams exhibit clear parallels in terms of their activity and size distribution with the soil samples. This finding highlights the potential for soil INPs to be transported into Arctic streams.

The INPs identified in Arctic streams provide additional insights into potential atmospheric sources of INPs. The high concentrations of INPs observed in the streams suggest that these waters might serve as a significant source of atmospheric INPs, especially given the observed variability in freezing onset temperatures and INP concentrations. Streams can act as conduits for transferring INPs from soil to the atmosphere, either through direct aerosolization caused by turbulence during high-flow conditions or by transporting these particles into the marine environment where they can subsequently aerosolize

through sea spray (Huang et al., 2021; Knackstedt et al., 2018; Raymond et al., 2012; Wieber et al., 2024b). Our findings indicate that streams with high INP concentrations could contribute to atmospheric INP levels, similar to observations in other freshwater systems (Larsen et al., 2017; Knackstedt et al., 2018). The presence of high INP concentrations in Arctic streams suggest their potential role as local sources of atmospheric INPs particularly in the context of Arctic amplification and increased freshwater discharge (Mankoff et al., 2020). However, further studies are needed to explore the linkage between these sources and aerosolized INPs, as well as their broader implications for cloud formation and regional climate.

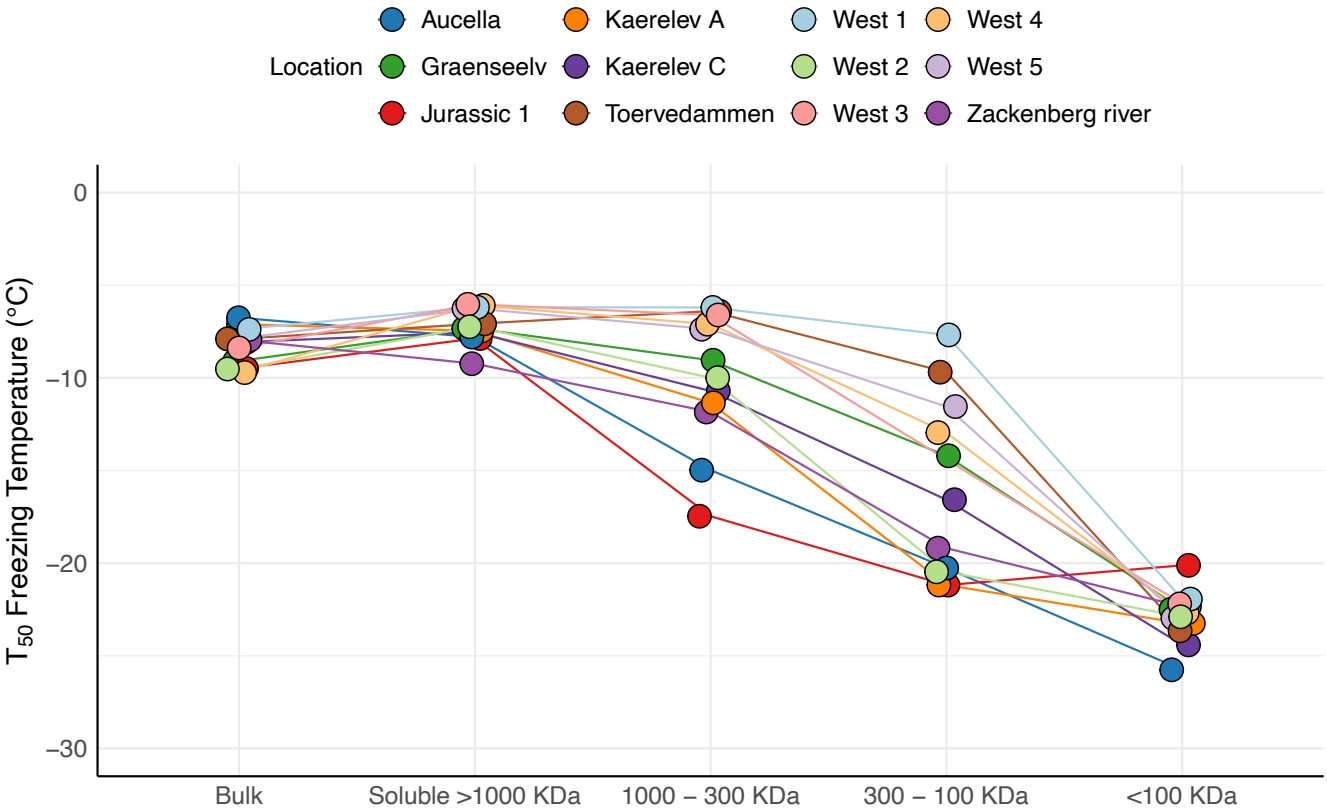

**Figure 7** illustrates the freezing activity at $T_{50}$ for water samples collected from twelve different locations in Northeast Greenland. The samples underwent various filtration treatments. The "Bulk" category represents the untreated sample. The "Soluble > 1000 kDa" category indicates samples that passed through a 0.2 μm filter but were retained on a 1000 kDa filter. The "1000-300 kDa" category comprises samples that passed through the 1000 kDa filter but were retained on the 300 kDa filter. The "300-100 kDa" category includes samples that passed through the 300 kDa filter but were retained on the 100 kDa filter. Finally, the "<100 kDa" category encompasses samples that passed through the 100 kDa filter.

## 3.5 Sources of INPs in Greenlandic streams

Terrestrial runoff has previously been identified as a major source of INPs in temperate rivers and lakes (Larsen et al., 2017; Knackstedt et al., 2018) and in the Arctic ocean (Irish et al., 2017; Irish et al., 2019; Wieber et al., 2024b). Larsen et al. (2018) found that most highly active INPs in the river Rhine were transported to the freshwater system through surface or subsurface water flow. The primary sources of INPs were identified as plant surfaces, plant litter, and soil, with an implicit indication of bacterial and fungal sources, respectively. Due to a tight coupling of INPs to seasonal changes in river discharge, the watershed of the Maumee River was identified as primary source of INPs to the river (Knackstedt et al., 2018).

If soil INPs are a major source of INPs in streams, we would expect to observe a positive correlation between soil and stream INP concentrations. Furthermore, in areas with steep terrain, less time is available for water to percolate through soil and accumulate INPs before entering the stream. Larger catchment areas, typically associated with longer streams, might also increase the potential for INP accumulation due to extended flow paths and surface interactions. Surprisingly, we did not observe significant correlations between INP$_{-10}$ concentrations in streams and catchment area size, or the slope of the terrain. Only a weak, non-significant correlation between soil and stream INP$_{-10}$ concentrations was found (R = 0.23, p > 0.05) (Supplementary Fig. 8). These findings suggest that while soil may contribute INPs to streams, other factors such as local hydrology, and dilution effects likely obscure a clear relationship. The onset freezing temperatures for the streams were in general lower compared to the adjacent soil samples (Supplementary Figs. 1 and 7). However, when comparing the T$_{50}$ freezing temperatures we found that the INPs in soil and stream were mainly in the same size range between 1000 and 300 KDa, which could fit with soluble fungal agglomerated INPs. This could suggest that terrestrial runoff is a potential source of INPs in the streams.

Precipitation is unlikely to be a significant source of INPs in the studied streams, as INP concentrations in snow are typically much lower than those measured in the streams. Previous studies report INP$_{-10}$ concentrations in snow ranging from as low as $1.2 \cdot 10^{-2}$ INP$_{-10}$ mL$^{-1}$. to approximately $8 \cdot 10^{1}$ INP$_{-10}$ mL$^{-1}$, which are significantly lower than the concentrations observed in stream water (Christner et al., 2008; Santl-Temkiv et al., 2019; Creamean et al., 2019; Brennan et al., 2020). This is supported by a Spearman's rank correlation analysis which showed no correlation between INP$_{-10}$ in the streams or soil and percentage of watershed snow cover (p > 0.05). This fits with previous studies of INPs in freshwater systems, where direct input to rivers from precipitation was found to be a minor contribution of INPs (Knackstedt et al., 2018; Moffett et al., 2018). Glacial outwash sediments were found to have a remarkably high nucleating ability (Tobo et al., 2019). Glacial outwash was shown to be the major freshwater runoff source in e.g., Nuup Kangerlua, dominating over rainfall and tundra which would imply that streams mostly receive their water from this runoff (Oksman et al., 2022). Glacial outwash could therefore be an important source of INPs in streams.

Finally, INPs could be produced by microorganisms that are autochthonous to the streams. Only a few studies have investigated INA organisms present in freshwater systems (Baloh et al., 2021; Benson et al., 2019) and different species of INA *Pseudomonas* have been found, which often grow in biofilms (Morris et al., 2007). Biofilms growing on stones, which could

be potential sources of autochthonous INPs in streams, are complex aggregations of algae, bacteria, and fungi (Pastor et al.,
2020; Battin et al., 2016). Morris et al. (2007) isolated 60-6000 *P. syringae* cells $g^{-1}$ wet weight of biofilm from river biofilms
at pristine settings characterized by low $NO_3^-$ concentrations (Morris et al., 2007). A study on biofilm growth in the Grenseelev
and Kaerelev, found that biofilm accrual was largely driven by high $NO_3^-$ concentrations in the stream (Pastor et al., 2020). In
our study, we observed a weak non-significant negative correlation between $INP_{-10}$ in the streams and $NO_3^-$, TDN, and DIN
concentrations (Supplementary Fig. 8), which points against biofilms as their predominant source. The size analysis of INPs
in the streams (Fig. 7) are consistent with presence of predominantly fungal-type INpro at most locations, which are typically
produced by soil fungi and could not be explained by the presence of INA bacteria in the stream biofilms. In addition, if a
majority of INPs were produced in the streams, we would expect that changes in water chemistry, reflecting biogeochemical
processes in the streams, would also correlate with changes in INP concentration. Aside from the $NO_3$, TDN, and DIN, we
observed no other correlations between $INP_{-10}$ and chemical parameters using Spearman's rank correlation (Supplementary
Fig. 8). Overall, we found no evidence for autochthonous production of INPs in the streams based on the parameters that we
recorded. To better understand the microbial transfer dynamics between soils and streams, future studies should include parallel
analyses of both soil and stream microbial communities. This would enable a more detailed assessment of microbial transfer,
particularly for ice-nucleating taxa, which is crucial for understanding the links between these two environments.

## 4 Conclusions

This study presents a detailed analysis of soil and freshwater INPs in High Arctic Greenland, offering critical insights into
their sources and diversity. The findings for the first time describe parallel measurements of INP concentrations in Arctic soil
and stream systems and incorporates microbial community analysis, providing direct connections between microbial taxa and
their respective INP contributions which opens the necessity for more studies investigating these environments. We found that
soil contained INPs that induced freezing at high temperatures, i.e., generally above -8°C. The $INP_{-10}$ concentrations varied by
2 orders of magnitude between locations, from $3.19 \cdot 10^4$ $g^{-1}$ to $1.55 \cdot 10^6$ $g^{-1}$. Additionally, using filtration through a series
of filters with decreasing cut-offs, we found that soil INPs at some locations were associated with soil particles or microbial
membranes, while at other locations they were present in the soluble fraction, likely excreted or detached from microbial
membranes. We found that *Mortierella*, a potential producer of INPs, was present across most locations and that this taxon
could be responsible for the production of these BioINPs. We took a novel approach using Spearman's rank correlations
between taxa relative abundance and INP concentration which provided new insights into potential producers of the warm
temperature INPs. Based on these correlations we hypothesize a diverse and hitherto overlooked number of bacterial genera
and fungi could produce these warm temperature INPs. To test this hypothesis the identified taxa, need to be isolated from
Arctic soil. Alternatively, representatives of the respective taxa that are stored in culture collections could be tested for their
INA.

In streams, INP concentrations were similar to those observed in temperate region rivers, such as the Mississippi and Gwaun Rivers. However, these concentrations were lower than those reported for other Arctic freshwater systems like thermokarst lakes. These concentrations demonstrated a positive but not significant correlation with INP concentrations in soil, which indicates that INPs are transported from soil into adjacent streams but are not the sole source for stream INPs.

Our findings, support the previously posed hypothesis that Arctic soil acts as an INP reservoir for streams, which could aerosolize directly from the soil or the streams themselves, or be transported further into the ocean, getting airborne through sea spray. In this way, the highly active INPs could impact cloud formation and climate, implying that bioINPs from soils and streams play a significant, yet complex, role in the Arctic climate system. Our findings underscore the importance of understanding the transport and potential atmospheric impact of bioINPs. While the direct quantification of aerosolization was beyond the scope of this study, future research should focus on deciphering the contributions from various sources, such as active layer soil, runoff, and marine emissions, combining the approaches used in this study with those employed in studies like Barry et al. (2023a, 2023b), to fully elucidate their roles in cloud formation and climate processes. As permafrost thaws and glaciers recede at an alarming rate, it has become increasingly important to understand the potential impact of these changes on INP concentrations in the Arctic. Understanding the origins and prevalence of INPs in this region, especially in the context of a warming climate, holds significance for accurate climate predictions. Thus, this research is an important contribution to understanding the dynamics of microbial communities and identifying the key contributors responsible for producing highly active INPs within Arctic soil microbial communities.

**Competing interests**

The authors declare that they have no conflict of interest.

**Author Contributions**

T.Š-T., L.C.L-H. K.F and A.P designed the research project. T.Š-T., L.C.L-H and L.Z.J. supervised sample collection. A.P. collected all samples. P.N. supervised the preparation of soil samples. J.K.S. and L.Z.J. prepared all samples and performed all ice nucleation measurements. L.Z.J performed the bioinformatic. C.P. conducted the grain size distribution analyses. L.Z.J. J.K.S, K.F and T.Š-T wrote the manuscript with contributions from all other coauthors.

**Financial support**

This work was supported by The Novo Nordisk Foundation (NNF19OC0056963), The Villum Foundation (23175, 28351, and 37435), The Danish National Research Foundation (DNRF106, to the Stellar Astrophysics Centre, Aarhus University), The Danish Agency for Higher Education and Science (1113-00025B), and The Carlsberg Foundation (CF21-0630). The project

was further developed in the frame of FACE-IT, which has received funding from the European Union's Horizon 2020 research and innovation program under grant agreement No. 869154.

## Acknowledgements

The authors are grateful to Egon R. Fransen for his logistic support. We would like to thank Britta Poulsen for excellent technical assistance with DNA extractions and sequencing. Thanks to Gunnar Rasmussen for technical help with the TC and TN measurements. We would also like to thank Corina Wieber for help and assistance with the ice nucleation set-up.

The authors thank Zackenberg logistics for assistance at the Zackenberg Research Station. We thank Núria Catalán and Cecilie M. H. Holmboe for help in sample collection.

## Data Availability

Ice nucleation data and soil geological data are available upon request.

Stream biogeochemical data and snowcover data are avalible at:

https://doi.pangaea.de/10.1594/PANGAEA.963212.

Sequence data obtained from this project are available at the European Nucleotide Archive under the accession number: PRJNA1137255

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
