# Peer review of "Linking Biogenic High-Temperature Ice Nucleating Particles in Arctic soils and Streams to Their Microbial Producers"

_Aerosol Research, 2024_

## Referee Comment (RC1)

Jensen et al. present a valuable study on INPs in soil and stream water samples and the microbial community composition in soil from multiple sites in Arctic Greenland. Their research includes size filtering to assess the different types of INPs in the samples. Compared to earlier studies, the INP concentrations in the soils were found to be somewhat lower. The authors also explore the potential linkage between INPs in soils and streams, aiming to test the hypothesis from previous research that soil-derived INPs may become airborne in the Arctic via the water-atmosphere interface. The authors conclude that the INPs detected in the soil are likely of fungal origin, specifically from species such as *Mortierella sp.,* and suggest that the INPs found in the streams may be linked to those in the soil. While the study is interesting and merits publication, several critical issues should be addressed prior to its acceptance.

**General comments:**

**"Permafrost" INPs:** My primary concern with this manuscript is the claim that permafrost samples were evaluated, which is inaccurate. The authors collected surface soil samples from the active layer, not permafrost. Active layer soil can differ significantly in composition from permafrost, as it is generally "younger" and largely composed of deposited loess. Thawed permafrost soil is typically located between the active layer and underlying frozen permafrost table, with the exception of coastal and freshwater shoreline erosion. The authors should avoid referring to these samples as permafrost, as this characterization is misleading.

**Blank correction:** The manuscript mentions the use of filtered Milli-Q water as a negative control, but were the samples blank-corrected using these spectra? It is essential to use the blanks to correct the spectra, given that Milli-Q water was involved in the sample preparation process.

**Spectra error bars:** In Figures 2, 3, 4, 6, 7, as well as in the INP spectra and aerosol size distribution figures in the SI, error or uncertainty bars should be shown.

**Comparison of onset temperatures:** Onset temperatures from different ice nucleation analytical techniques cannot be directly compared due to subtle

differences that may affect the detection limits of each instrument. For instance, nanoliter-sized droplets have a lower detection limit compared to microliter- or milliliter-sized droplets (see Figure 4 in Tobo, 2016). The authors should avoid comparing onset temperatures with those reported in previous studies throughout the manuscript.

**INP sizes:** The conclusion that INPs were either bound to soil particles or microbial membranes at certain locations, while other sites displayed a variety of soluble INPs with different molecular sizes, is particularly interesting given the heterogeneity across sites. In the results, the authors suggest that fungal INpro are the most likely candidates (based on the discussion starting at line 297). However, are there other possible materials that could serve as INPs? For example, carbohydrates (polysaccharides) can range from 100-1000 kDa, although it's unclear if these sizes are typically found in soils or streams, and they have been shown to nucleate ice (e.g., Alpert et al., 2022). Furthermore, considering the sieving and comminution using a mortar, could cellular material have fragmented into smaller pieces? How easily do these proteins detach from cell walls? It would be advantageous for the authors to rule out other potential ice-nucleating materials based on size to support their claim that these particles are most likely INpro, either on their own or attached to soil particle surfaces.

**Conclusions not supported by results:** Some conclusions in the manuscript are not fully supported by the results. For instance, on lines 515-516, the authors state, "The findings for the first time describe parallel measurements of INP concentrations in Arctic soil and stream systems and open the necessity for more studies investigating these environments." However, this is not technically the first time such measurements have been made, as Barry et al. (2023a) compared freshwater outflow INPs with soil INPs.

On lines 463-464, the authors claim, "The presence of high INP concentrations in Arctic streams has implications for cloud formation and regional climate," and in the conclusions, they assert that "In this way, the highly active INPs could impact cloud formation and climate, implying that bioINPs from soils and streams play a significant, yet complex, role in the Arctic climate system." This conclusion is

somewhat overstated, given the current evidence, especially since INPs in aerosols were not measured or linked to the soil and stream water source samples. The authors should avoid such claims and instead focus the intent on the characterization of potential local Arctic sources of INPs.

Additionally, the statement in the conclusions, "Stream INP concentrations demonstrated a positive but not significant correlation with INP concentrations in soil, which indicates that INPs are transported from soil into adjacent streams but are not the sole source for stream INPs," raises questions. Why were 16S and ITS analyses not performed for the stream water samples? Without this information, it is challenging to draw meaningful connections between the soil as a source of INPs and the processes that facilitate their transfer to streams.

**Specific comments:**

Line 47: "…ice nuclei to form ice particles…" should be "ice nucleating particles to form ice crystals…"

Lines 47-48: This statement is inaccurate. Interest in bioINPs dates back to the 1970s, with pioneering studies by Schnell and Vali (1976) and Vali (1976). The authors should acknowledge these foundational works. Additionally, more recent reviews, such as Huang et al. (2021), should be cited in this context.

Line 49: It would be best to update to the newest IPCC report.

Lines 74-75: "Ice nucleation below -15°C is initiated by abiotic INPs…" and ""…while the only known INPs that are active above -15°C and present at relevant concentrations are of biotic origin…" are both inaccurate statements. See Kanji et al. (2017) and Murray et al. (2012). Certain minerals have been shown to nucleate ice above -15°C, although in low concentrations (e.g., Harrison et al. (2019)).

Lines 76-77: This statement on the types of bioINPs should be cited.

Line 77: This statement exhibits significant self-citation. The authors should consider incorporating several key papers on Arctic bioINPs, such as Bigg (1996), Bigg and Leck (2001), Creamean et al. (2022), Hartmann et al. (2020, 2021), Ickes et al. (2020a, b), Jayaweera and Flanagan (1982), Porter et al. (2022), etc. to provide a more comprehensive perspective. Some of these could also be used for the statement on lines 78-79.

Lines 89-90: The statement, "Aerosolization of INP by bubble bursting in freshwater bodies is more likely than in the ocean since more bubbles are produced by frequent small waves…" overlooks other factors like fetch and salinity that influence bubble concentration. Studies such as Cartmill and Yang (1993) have found higher bubble concentrations in saltwater, and Zinke et al. (2022) observed lower particle number fluxes in fresher water compared to saltier water. These factors should be considered for a more accurate interpretation.

Line 107: What is the classification of the underlying permafrost (e.g., thick, continuous, discontinuous)? Additionally, was the ground completely free of snow and ice? More details about the sites are useful for context.

Line 121: When referring to airflow, do the authors mean clean air or dry nitrogen? Additionally, are there any concerns regarding the evaporation of the droplets at a sheath flow rate of 15-20 lpm?

Lines 137-139: Why were the samples freeze-dried overnight? Could the use of a desiccator potentially stress the microbial cells in the samples, possibly affecting their viability? Additionally, does the mortaring process lead to any degradation of the INPs in the samples? Finally, what is the rationale behind sieving the samples? It is unclear why this preparation method was chosen over simply freezing, suspending, and testing the soil samples. Although the authors cite Conen and Yakutin (2018), their methodology differed slightly, as they used air drying rather than freeze-drying and did not employ a mortar. An explanation justifying the chosen steps in this study would be helpful.

Fig 3: This is a well-constructed summary figure; however, could the authors also include the other figures referenced on line 237 (Creamean et al., 2020; Schnell and Vali, 1976)? Incorporating these would provide a more comprehensive overview.

Line 242: Since the authors note that the Tobo et al. study focused on glacial outwash sediment, it would be helpful to specify that the Barry et al. study pertains to permafrost.

Lines 243-244: How do these TC values compare to others in the literature for similar soils?

Lines 245-246: More biomass and soil carbon content than what?

Lines 251-253: Other factors contributing to the large variations may stem from the sample preparation methods. Conen et al. used sieving but did not employ mortaring, while Tobo et al. utilized neither technique. While differences in microbial community composition or soil properties could influence the results, the impact of the varying sample preparation methods should not be overlooked.

Lines 254-255: While Santl-Temkiv et al. provides a valuable review on aerobiology, the authors should include other relevant papers as mentioned in previous comments.

Fig 3: The spectra from Conen et al. are somewhat difficult to distinguish. I recommend using a different color for clarity. Additionally, it would be beneficial to assign different colors and/or markers to the spectra from the 2011 and 2018 studies for better differentiation.

Lines 271-277: This text would be better placed in the methods section.

Figs 4 (and 7): Technically, the sample should not be labeled as "bulk" as indicated on the x-axis. The authors should refer to this as "≤ 63 μm" instead.

Line 287: This analysis focuses on INP size and inferred composition, rather than direct composition. Additionally, it would be helpful to mention whether other studies, such as Barry et al. (2023a, b), observe significant variations in the INPs present in soil. The Barry et al. studies investigated concentrations and the effects of heat and peroxide treatments on composition (2023a and b), along with size filtering (2023b only).

Line 304-205: The statement, "The gradual loss of INA during filtration at the different locations suggests a mixture of different-sized INPs, predominantly originating from fungi," needs clarification. Where is this information presented in the manuscript, or what other evidence supports this claim?

Lines 308-318: This is a nice summary, but would fit better in the conclusions section.

Line 316: Regarding the "upward fluxes," was the surface marshy or dry? Positive fluxes from the surface would depend on the surface aridity. This is an example of how describing the landscape of the sampling locations would be beneficial. Additionally, on line 390, wind erosion is mentioned; however, this also depends on surface aridity, which may not be realistic if the sampling locations were marshy.

Lines 337-344: The authors conclude that the abundances of known INP-producing species are very low for both 16S and ITS, with only sequences affiliated with *Acremonium* (at one location) and *Mortierella* (at most locations) present in their dataset. They suggest that the observed taxa might be INP producers that have not yet been recognized as such. However, could the INPs be derived from other organic materials, rather than exclusively from cellular or proteinaceous sources?

Lines 417-445: The authors should summarize and directly compare their findings to those of Barry et al. (2023a), as their INP results are derived from freshwater thermokarst lakes in the Arctic and are likely the most relevant for comparison with the Arctic stream water analyzed in this study. Barry et al. also included comparisons with locally sampled permafrost and active layer soils, while the other studies cited are focused on temperate regions.

Lines 425-427: Huang et al. (2021) discuss how local Arctic sources can be rich in INPs,

yet the concentration of aerosol INPs remains low. Given this context, is this finding truly surprising? Additionally, on lines 527-528, the authors state that "In streams, INP concentrations defied conventional expectations, exhibiting elevated concentrations contrary to the typical decrease towards polar regions." Is this assertion accurate?

Lines 434-445: If these are all possible explanations, why would they apply specifically to the stream samples and not to the soil? This suggests that the INP populations in the two environments are not the same.

Lines 487-489: Missing some key references here that looked at INPs in snowmelt, such as Brennan et al. (2020), Creamean et al. (2019), Stopelli et al. (2015, 2017). It would be useful to compare values to more than just Christner et al., (2008) and Santl-Temkiv et al. (2018).

Lines 536-537: The statement, "…future research should focus on deciphering the contributions from various sources such as soil, runoff, and marine emissions to fully elucidate their roles in cloud formation and climate processes," should acknowledge the work of Barry et al. (2023a), who investigated a wide range of potential sources, including those mentioned, and linked their findings to INP data collected upwind and downwind of thermokarst lakes. They should receive appropriate credit for their contributions in this context.

Supplemental Figs 3 and 4: These figures seem central to the main takeaways, why are they not shown in the main text?

**References:**

Alpert, P. A., Kilthau, W. P., O'Brien, R. E., Moffet, R. C., Gilles, M. K., Wang, B., Laskin, A., Aller, J. Y., and Knopf, D. A.: Ice-nucleating agents in sea spray aerosol identified and quantified with a holistic multimodal freezing model, Science Advances, 8, eabq6842, https://doi.org/10.1126/sciadv.abq6842, 2022.

Barry, K. R., Hill, T. C. J., Nieto-Caballero, M., Douglas, T. A., Kreidenweis, S. M., DeMott, P. J., and Creamean, J. M.: Active thermokarst regions contain rich sources of

ice-nucleating particles, Atmospheric Chemistry and Physics, 23, 15783–15793, https://doi.org/10.5194/acp-23-15783-2023, 2023a.

Barry, K. R., Hill, T. C. J., Moore, K. A., Douglas, T. A., Kreidenweis, S. M., DeMott, P. J., and Creamean, J. M.: Persistence and Potential Atmospheric Ramifications of Ice-Nucleating Particles Released from Thawing Permafrost, Environ. Sci. Technol., 57, 3505–3515, https://doi.org/10.1021/acs.est.2c06530, 2023b.

Bigg, E. K.: Ice forming nuclei in the high Arctic, 1996.

Bigg, E. K. and Leck, C.: Cloud-active particles over the central Arctic Ocean, J. Geophys. Res., 106, 32155–32166, https://doi.org/10.1029/1999JD901152, 2001.

Brennan, K. P., David, R. O., and Borduas-Dedekind, N.: Spatial and temporal variability in the ice-nucleating ability of alpine snowmelt and extension to frozen cloud fraction, Atmospheric Chemistry and Physics, 20, 163–180, https://doi.org/10.5194/acp-20-163-2020, 2020.

Cartmill, J. W. and Yang Su, M.: Bubble size distribution under saltwater and freshwater breaking waves, Dynamics of Atmospheres and Oceans, 20, 25–31, https://doi.org/10.1016/0377-0265(93)90046-A, 1993.

Creamean, J. M., Mignani, C., Bukowiecki, N., and Conen, F.: Using freezing spectra characteristics to identify ice-nucleating particle populations during the winter in the Alps, Atmospheric Chemistry and Physics, 19, 8123–8140, https://doi.org/10.5194/acp-19-8123-2019, 2019.

Creamean, J. M., Barry, K., Hill, T. C. J., Hume, C., DeMott, P. J., Shupe, M. D., Dahlke, S., Willmes, S., Schmale, J., Beck, I., Hoppe, C. J. M., Fong, A., Chamberlain, E., Bowman, J., Scharien, R., and Persson, O.: Annual cycle observations of aerosols capable of ice

formation in central Arctic clouds, Nat Commun, 13, 3537, https://doi.org/10.1038/s41467-022-31182-x, 2022.

Harrison, A. D., Lever, K., Sanchez-Marroquin, A., Holden, M. A., Whale, T. F., Tarn, M. D., McQuaid, J. B., and Murray, B. J.: The ice-nucleating ability of quartz immersed in water and its atmospheric importance compared to K-feldspar, Atmospheric Chemistry and Physics, 19, 11343–11361, https://doi.org/10.5194/acp-19-11343-2019, 2019.

Hartmann, M., Adachi, K., Eppers, O., Haas, C., Herber, A., Holzinger, R., Hünerbein, A., Jäkel, E., Jentzsch, C., Pinxteren, M., Wex, H., Willmes, S., and Stratmann, F.: Wintertime Airborne Measurements of Ice Nucleating Particles in the High Arctic: A Hint to a Marine, Biogenic Source for Ice Nucleating Particles, Geophys. Res. Lett., 47, https://doi.org/10.1029/2020GL087770, 2020.

Hartmann, M., Gong, X., Kecorius, S., Van Pinxteren, M., Vogl, T., Welti, A., Wex, H., Zeppenfeld, S., Herrmann, H., Wiedensohler, A., and Stratmann, F.: Terrestrial or marine – indications towards the origin of ice-nucleating particles during melt season in the European Arctic up to 83.7° N, Atmos. Chem. Phys., 21, 11613–11636, https://doi.org/10.5194/acp-21-11613-2021, 2021.

Ickes, L., Porter, G. C. E., Wagner, R., Adams, M. P., Bierbauer, S., Bertram, A. K., Bilde, M., Christiansen, S., Ekman, A. M. L., Gorokhova, E., Höhler, K., Kiselev, A. A., Leck, C., Möhler, O., Murray, B. J., Schiebel, T., Ullrich, R., and Salter, M.: Arctic marine ice nucleating aerosol: a laboratory study of microlayer samples and algal cultures, Aerosols/Laboratory Studies/Troposphere/Physics (physical properties and processes), https://doi.org/10.5194/acp-2020-246, 2020a.

Ickes, L., Porter, G. C. E., Wagner, R., Adams, M. P., Bierbauer, S., Bertram, A. K., Bilde, M., Christiansen, S., Ekman, A. M. L., Gorokhova, E., Höhler, K., Kiselev, A. A., Leck, C., Möhler, O., Murray, B. J., Schiebel, T., Ullrich, R., and Salter, M. E.: The ice-nucleating activity of

Arctic sea surface microlayer samples and marine algal cultures, Atmos. Chem. Phys., 20, 11089–11117, https://doi.org/10.5194/acp-20-11089-2020, 2020b.

Jayaweera, K. and Flanagan, P.: Investigations on biogenic ice nuclei in the Arctic atmosphere, Geophysical Research Letters, 9, 94–97, https://doi.org/10.1029/GL009i001p00094, 1982.

Kanji, Z. A., Ladino, L. A., Wex, H., Boose, Y., Burkert-Kohn, M., Cziczo, D. J., and Krämer, M.: Overview of Ice Nucleating Particles, https://doi.org/10.1175/AMSMONOGRAPHS-D-16-0006.1, 2017.

Murray, B. J., O'Sullivan, D., Atkinson, J. D., and Webb, M. E.: Ice nucleation by particles immersed in supercooled cloud droplets, Chem. Soc. Rev., 41, 6519–6554, https://doi.org/10.1039/C2CS35200A, 2012.

Porter, G. C. E., Adams, M. P., Brooks, I. M., Ickes, L., Karlsson, L., Leck, C., Salter, M. E., Schmale, J., Siegel, K., Sikora, S. N. F., Tarn, M. D., Vüllers, J., Wernli, H., Zieger, P., Zinke, J., and Murray, B. J.: Highly Active Ice-Nucleating Particles at the Summer North Pole, Journal of Geophysical Research: Atmospheres, 127, e2021JD036059, https://doi.org/10.1029/2021JD036059, 2022.

Schnell, R. C. and Vali, G.: Biogenic Ice Nuclei: Part I. Terrestrial and Marine Sources, J. Atmos. Sci., 33, 1554–1564, 1976.

Stopelli, E., Conen, F., Morris, C. E., Herrmann, E., Bukowiecki, N., and Alewell, C.: Ice nucleation active particles are efficiently removed by precipitating clouds, Sci Rep, 5, 16433, https://doi.org/10.1038/srep16433, 2015.

Stopelli, E., Conen, F., Guilbaud, C., Zopfi, J., Alewell, C., and Morris, C. E.: Ice nucleators, bacterial cells and Pseudomonas syringae; in precipitation at Jungfraujoch, Biogeosciences, 14, 1189–1196, https://doi.org/10.5194/bg-14-1189-2017, 2017.

Tobo, Y.: An improved approach for measuring immersion freezing in large droplets over a wide temperature range, Sci Rep, 6, 32930, https://doi.org/10.1038/srep32930, 2016.

Vali, G., Christensen, M., Fresh, R. W., Galyan, E. L., Maki, L. R., and Schnell, R. C.: Biogenic Ice Nuclei. Part II: Bacterial Sources, Journal of the Atmospheric Sciences, 33, 1565–1570, https://doi.org/10.1175/1520-0469(1976)033<1565:BINPIB>2.0.CO;2, 1976.

Zinke, J., Nilsson, E. D., Zieger, P., and Salter, M. E.: The Effect of Seawater Salinity and Seawater Temperature on Sea Salt Aerosol Production, JGR Atmospheres, 127, https://doi.org/10.1029/2021JD036005, 2022.

---

## Author Comment (AC1)

Jensen et al. present a valuable study on INPs in soil and stream water samples and the microbial community composition in soil from multiple sites in Arctic Greenland. Their research includes size filtering to assess the different types of INPs in the samples. Compared to earlier studies, the INP concentrations in the soils were found to be somewhat lower. The authors also explore the potential linkage between INPs in soils and streams, aiming to test the hypothesis from previous research that soil-derived INPs may become airborne in the Arctic via the water-atmosphere interface. The authors conclude that the INPs detected in the soil are likely of fungal origin, specifically from species such as Mortierella sp., and suggest that the INPs found in the streams may be linked to those in the soil. While the study is interesting and merits publication, several critical issues should be addressed prior to its acceptance.

**General comments:**

**"Permafrost" INPs:** My primary concern with this manuscript is the claim that permafrost samples were evaluated, which is inaccurate. The authors collected surface soil samples from the active layer, not permafrost. Active layer soil can differ significantly in composition from permafrost, as it is generally "younger" and largely composed of deposited loess. Thawed permafrost soil is typically located between the active layer and underlying frozen permafrost table, with the exception of coastal and freshwater shoreline erosion. The authors should avoid referring to these samples as permafrost, as this characterization is misleading.

Thank you for highlighting this important distinction. We acknowledge that the samples we collected were from the active layer, not from permafrost itself, and we regret any confusion caused by this mischaracterization. We have carefully revised the manuscript to refer to these samples as "active layer soil" throughout the text to ensure accurate representation. We appreciate your observation

**Blank correction:** The manuscript mentions the use of filtered Milli-Q water as a negative control, but were the samples blank-corrected using these spectra? It is essential to use the blanks to correct the spectra, given that Milli-Q water was involved in the sample preparation process.

Thank you for your comment. We understand the importance of blank corrections, especially given that Milli-Q water was used during sample preparation. Milli-Q water was used as a negative control, and its freezing behavior has been added to the fraction frozen plots (Supplementary Figures 1 and 7). These controls consistently show onset freezing temperatures below -15°C, which do not overlap with the temperature range of primary interest (0°C to -15°C).
Given this clear separation, background corrections were deemed unnecessary for this study. However, we acknowledge that applying such corrections may be crucial in cases where there is a significant overlap between control and sample data.

**Spectra error bars:** In Figures 2, 3, 4, 6, 7, as well as in the INP spectra and aerosol size distribution figures in the SI, error or uncertainty bars should be shown.

Thank you for your comment. We appreciate your suggestion to include error or uncertainty bars. The INP spectra presented in Figures 2, 3, 4, 6, and 7 does not involve binning the data into temperature intervals, as we rely on cumulative freezing spectra derived from individual droplets or wells. This approach provides precise freezing temperatures for each droplet, without aggregating the data into bins. Consequently, there is no statistical variation across temperature intervals to calculate error bars. Furthermore, each sample is analyzed using 80 droplets, instead of analyzing smaller numbers of droplets in replicates, as has been recommended by Polen et al (2018) (Polen et al., 2018). This also means a high coverage of the freezing temperature distribution within the sample. This high number of droplets allows us to capture the variability of freezing events across the sample population with high confidence, minimizing the need for additional statistical representation such as error bars.

Regarding the size distribution of soil particles included in the Supplementary Information, these data are based on four independent measurements, each obtained from different sampling sites. These samples are used to investigate potential differences between sites and not merely to provide an average for the locations. Hence, we believe that it would be appropriate to keep the measurements separate.
We hope this explanation addresses your concerns, but we are happy to provide further clarification if needed.

**Comparison of onset temperatures:** Onset temperatures from different ice nucleation analytical techniques cannot be directly compared due to subtle differences that may affect the detection limits of each instrument. For instance, nanoliter-sized droplets have a lower detection limit compared to microliter- or milliliter-sized droplets (see Figure 4 in Tobo, 2016). The authors should avoid comparing onset temperatures with those reported in previous studies throughout the manuscript.

We agree with this concern regarding the comparability of onset freezing temperatures across studies using different droplet volumes or different input amount of soils in freezing assays. We have revised the manuscript to explicitly discuss the potential implications of methodological differences. Specifically, we added a section that acknowledges that larger droplet volumes are more sensitive to detecting rarer INPs. Many of the studies we compare our results to, used larger volumes and therefore their sensitivity is higher than in our assays, supporting the statement that INPs observed in our study indded show a higher onset temperature than other studies. In these revisions, we now emphasize both the methodological context and the unique INP activity observed in the Arctic soils we studied. By addressing these points, we believe the discussion now provides a balanced interpretation of the results while maintaining scientific rigor.

*"Our results showed higher onset temperatures (between -1.5 °C and -4.7 °C) compared to previous studies of Arctic soils (Fig. 3). Methodological differences, such as droplet volume used in freezing assays, must be considered when interpreting this trend. Studies using smaller volumes (e.g. 5 µL in Tobo et al., 2019), have a lower sensitivy and cannot be directly compared to our study. However, several studies used larger droplet volumes*

*(e.g., 50 µL in Conen et al., 2018; Barry et al., 2023b, and 100 µL in Conen et al., 2012), which have a higher sensitivity than the micro-Pinguin asssay and a comparable potential to detect rare highly active INPs. Therefore, the higher freezeing onset that we observe does not seem to only be linked to methodological differences but reflects differences in the INP popoluations in these environments. INPs active at such high temperatures are generally proteinaceous (Santl-Temkiv et al., 2022) and are often associated with microbial sources, including bacteria and fungi (Barry et al., 2023b; Tobo et al., 2019; Conen et al., 2011). The presence of higher onset temperatures in this study may indicate differences in either the identity or in the activity of their microbial producers across Arctic terrestrial environments."*

**INP sizes:** The conclusion that INPs were either bound to soil particles or microbial membranes at certain locations, while other sites displayed a variety of soluble INPs with different molecular sizes, is particularly interesting given the heterogeneity across sites. In the results, the authors suggest that fungal INpro are the most likely candidates (based on the discussion starting at line 297). However, are there other possible materials that could serve as INPs? For example, carbohydrates (polysaccharides) can range from 100-1000 kDa, although it's unclear if these sizes are typically found in soils or streams, and they have been shown to nucleate ice (e.g., Alpert et al., 2022). Furthermore, considering the sieving and comminution using a mortar, could cellular material have fragmented into smaller pieces? How easily do these proteins detach from cell walls? It would be advantageous for the authors to rule out other potential ice-nucleating materials based on size to support their claim that these particles are most likely INpro, either on their own or attached to soil particle surfaces.

Thank you for your insightful question. Studies of defined carbohydrates, such as cellulose and lignin have shown that these nucleate ice at temperatures below -13°C (Bogler and Borduas-Dedekind, 2020; Hiranuma et al., 2015), while fungal and bacterial proteins nucleate ice at temperatures between -1 and 13°C (Schwidetzky et al., 2023; Hartmann et al., 2022). The freezing profiles we recorded are thus closer to known proteinaceous INPs than with those of carbohydrates. Alpert et al., (2022), who studied algal exudates, consisting of a mixture of proteinaceous and polysaccharidic compounds and found that their activity is typically lower than that of the terrestrial proteinaceous INPs active above -13°C.

While we cannot entirely rule out the possibility that some INPs might have been detached from the cells during the mortaring process, it is important to note that mortaring was conducted as a standard soil preparation step prior to sieving. This process was essential for breaking down soil aggregates that had formed during freeze-drying, ensuring effective sieving and sample consistency. Given that the mortaring was not performed with a high intensity to specifically degrade cells or INPs, we do not expect significant degradation or loss of INPs within the samples.

**Conclusions not supported by results:** Some conclusions in the manuscript are not

fully supported by the results. For instance, on lines 515-516, the authors state, "The findings for the first time describe parallel measurements of INP concentrations in Arctic soil and stream systems and open the necessity for more studies investigating these environments." However, this is not technically the first time such measurements have been made, as Barry et al. (2023a) compared freshwater outflow INPs with soil INPs.

We appreciate the reviewer's feedback and would like to address the concern regarding our statement on lines 515-516, where we refer to our study as the "first time" describing parallel measurements of INP concentrations in Arctic soil and stream systems. While we acknowledge that Barry et al. (2023a) surveyed potential INP sources in permafrost and adjacent water bodies in active thermokarst regions, we believe the distinction between our work and theirs warrants clarification. Barry et al. focused on the broader permafrost zone, specifically analyzing water from thermocast lakes, lagoons, and the ocean, while our study specifically investigates streams and adjacent soils. Furthermore, our study uniquely incorporates microbial community analysis, providing direct connections between microbial taxa and their respective INP contributions - an aspect that was not explored in Barry et al.'s work. Barry et al. did not perform microbial sequencing, and thus their findings did not delve into the potential biological sources of INPs at the level we present in our manuscript.
We have further added this to the manuscript:

*"This study presents a detailed analysis of soil and freshwater INPs in High Arctic Greenland, offering critical insights into their sources and diversity. The findings for the first time describe parallel measurements of INP concentrations in Arctic soil and stream systems and incorporates microbial community analysis, providing direct connections between microbial taxa and their respective INP contributions which opens the necessity for more studies investigating these environments."*

On lines 463-464, the authors claim, "The presence of high INP concentrations in Arctic streams has implications for cloud formation and regional climate," and in the conclusions, they assert that "In this way, the highly active INPs could impact cloud formation and climate, implying that bioINPs from soils and streams play a significant, yet complex, role in the Arctic climate system." This conclusion is somewhat overstated, given the current evidence, especially since INPs in aerosols were not measured or linked to the soil and stream water source samples. The authors should avoid such claims and instead focus the intent on the characterization of potential local Arctic sources of INPs.

We appreciate the reviewer's concern and agree that, given the current scope of our study, the conclusions may be overstated, particularly as we did not measure or directly link aerosolized INPs to the soil and stream water samples. In response, we have revised the manuscript to tone down the language and focus on the characterization of potential local Arctic sources of INPs. We now emphasize the observed high INP concentrations in Arctic streams as a significant finding in understanding the potential role of these

freshwater systems as contributors to atmospheric INPs, without overreaching into claims about their broader impact on cloud formation and climate.

*"Our findings indicate that streams with high INP concentrations could contribute to atmospheric INP levels, similar to observations in other freshwater systems (Larsen et al., 2017; Knackstedt et al., 2018). The presence of high INP concentrations in Arctic streams suggest their potential role as local sources of atmospheric INPs particularly in the context of Arctic amplification and increased freshwater discharge (Mankoff et al., 2020). However, further studies are needed to explore the linkage between these sources and aerosolized INPs, as well as their broader implications for cloud formation and regional climate. "*

Additionally, the statement in the conclusions, "Stream INP concentrations demonstrated a positive but not significant correlation with INP concentrations in soil, which indicates that INPs are transported from soil into adjacent streams but are not the sole source for stream INPs," raises questions. Why were 16S and ITS analyses not performed for the stream water samples? Without this information, it is challenging to draw meaningful connections between the soil as a source of INPs and the processes that facilitate their transfer to streams.

We appreciate the reviewer's insightful comment. We fully agree that performing 16S, ITS, 18S, or metagenomic analyses on stream water samples would have provided valuable insights into the microbial community in the streams. Ideally, we would have included these analyses in this study. However, due to logistical constraints, we were unable to collect stream samples for DNA analysis.
That said, we believe meaningful conclusions can still be drawn based on the data we have. Our results, which show the presence of INPs in both soils and streams, along with their size distribution and activity, suggest a transfer of bioINPs from the soil to the stream environment. Although we were not able to directly identify the microbial contributors in the streams, the positive correlation (albeit not significant) between INP concentrations in soil and streams supports the idea that soils are a potential source for stream INPs. Additionally, the size distribution patterns observed in both environments suggest that microorganisms which produced these INPs were related and that similar processes influenced the INPs size distribution in both soils and streams, further supporting the possibility of bioINP transfer between these two environments.
Moreover, it is important to note that the INPs transferred to the streams are predominantly in the soluble fraction, as suggested by our filtration experiments. This could indicate a decoupling between the microbial producers and the actual INPs present in the stream environment. The processes facilitating this decoupling, such as cell lysis or the release of extracellular INPs, may further obscure the direct link between soil microbial communities and INPs in the streams.
While identifying the exact microbial sources in the streams would have been ideal, we believe our conclusions remain valid and meaningful based on the available data. We acknowledge this as a limitation of the study

**Specific comments:**

Line 47: "…ice nuclei to form ice particles…" should be "ice nucleating particles to form ice crystals…"

Thank you for this comment, which has been implemented in the new version of the manuscript.

Lines 47-48: This statement is inaccurate. Interest in bioINPs dates back to the 1970s, with pioneering studies by Schnell and Vali (1976) and Vali (1976). The authors should acknowledge these foundational works. Additionally, more recent reviews, such as Huang et al. (2021), should be cited in this context.

Thank you for this comment, which has been implemented in the new version of the manuscript.

Line 49: It would be best to update to the newest IPCC report.

Thank you for this comment, which has been implemented in the new version of the manuscript.

Lines 74-75: "Ice nucleation below -15°C is initiated by abiotic INPs…" and ""…while the only known INPs that are active above -15°C and present at relevant concentrations are of biotic origin…" are both inaccurate statements. See Kanji et al. (2017) and Murray et al. (2012). Certain minerals have been shown to nucleate ice above -15°C, although in low concentrations (e.g., Harrison et al. (2019)).

We have revised the wording to addresses your concern.

*"Ice nucleation below -13°C can be initiated by abiotic INPs, such as mineral particles, soot, or by incidental INA from biomolecules, such as carbohydrates (Kinney et al., 2024). In contrast, proteinaceous INPs are the predominant type of INP that are active above -13°C, shown by both laboratory studies and in-situ measurements involving inactivation by heating (Murray et al., 2012; Cornwell et al., 2023; Kanji et al., 2017; Daily et al., 2022)."*

Lines 76-77: This statement on the types of bioINPs should be cited.

Thank you for this comment, which has been implemented in the new version of the manuscript.

Line 77: This statement exhibits significant self-citation. The authors should consider incorporating several key papers on Arctic bioINPs, such as Bigg (1996), Bigg and Leck (2001), Creamean et al. (2022), Hartmann et al. (2020, 2021), Ickes et al. (2020a, b), Jayaweera and Flanagan (1982), Porter et al. (2022), etc. to provide a more comprehensive perspective. Some of these could also be used for the statement on

lines 78-79.

Thank you for this comment, which has been implemented in the new version of the manuscript.

Lines 89-90: The statement, "Aerosolization of INP by bubble bursting in freshwater bodies is more likely than in the ocean since more bubbles are produced by frequent small waves..." overlooks other factors like fetch and salinity that influence bubble concentration. Studies such as Cartmill and Yang (1993) have found higher bubble concentrations in saltwater, and Zinke et al. (2022) observed lower particle number fluxes in fresher water compared to saltier water. These factors should be considered for a more accurate interpretation.

We appreciate the reviewer's insightful comment highlighting the complexity of factors influencing bubble concentrations and aerosolization processes in freshwater and saltwater environments. Upon further consideration, we have decided to remove this comparison from the manuscript. Since no quantitative measurements were conducted in this study to support a detailed comparison, we concluded that this discussion is not essential to the manuscript's focus.

Line 107: What is the classification of the underlying permafrost (e.g., thick, continuous, discontinuous)? Additionally, was the ground completely free of snow and ice? More details about the sites are useful for context.

Thank you for this comment. We have now added a brief section in the methods about the sampling site, along with relevant references. Unfortunately, we do not know whether the permafrost at the collection sites was thick, continuous, or discontinuous.

*"The study streams are located in the Northeast Greenland National Park, near the Zackenberg Research Station (74°28'N, 20°34'W). Streamflow in the region is predominantly derived from melting snow and glaciers, with additional contributions varying by stream. For example, Kærelv A, Kærelv C, Grænseelv, and West 1 receive water from small, seasonal snow patches, while Aucella, and Jurassic1 is partly fed by larger ice aprons adhered to mountainsides (Docherty et al., 2019; Hasholt and Hagedorn, 2000). This region has a polar tundra climate and is underlain by continuous permafrost with an active layer thickness of 0.4 - 0.8 m (Christiansen et al., 2008; Hollesen et al., 2011). Geologically, the area is divided into crystalline complexes to the west and sedimentary successions to the east, with Quaternary sediments covering the valley floor and slopes. For a broader overview of the region's climate, geology, and vegetation, see Riis et al. (2023)."*

Line 121: When referring to airflow, do the authors mean clean air or dry nitrogen? Additionally, are there any concerns regarding the evaporation of the droplets at a sheath flow rate of 15-20 lpm?

We thank the reviewer for this comment. We have now specified that the air supplied was HEPA filtered clean air, and that the tower first was flushed for 5 minutes at a constant flow rate of 20 LPM, while turning down the flow during the experiment.

*"The tower was first flushed with HEPA filtered clean for 5 minutes at a constant flow of 20 L/min before. Then, the samples were cooled at 1K min$^{-1}$ down to -30°C while supplying a constant HEPA filtered clean airflow between 5-10 L/min to keep the relative humidity low, avoiding condensation."*

Evaporation of the droplets due to airflow was tested and was insignificant.

Lines 137-139: Why were the samples freeze-dried overnight? Could the use of a desiccator potentially stress the microbial cells in the samples, possibly affecting their viability? Additionally, does the mortaring process lead to any degradation of the INPs in the samples? Finally, what is the rationale behind sieving the samples? It is unclear why this preparation method was chosen over simply freezing, suspending, and testing the soil samples. Although the authors cite Conen and Yakutin (2018), their methodology differed slightly, as they used air drying rather than freeze-drying and did not employ a mortar. An explanation justifying the chosen steps in this study would be helpful.

We appreciate the reviewer's concerns. Below, we provide clarification and rationale for each of the points raised.
1. **Freeze-Drying vs. Air Drying**:
   The decision to freeze-dry the samples was based on our aim to minimize the time required for soil drying, as opposed to air-drying used by Conen and Yakutin (2018). Prolonged air-drying might support microbial activity during its initial stages when water availability is still high, potentially leading to the production or degradation of INPs, which could alter the ice nucleation properties of the samples. Freeze-drying ensured rapid drying, thereby preserving the original INP content and composition.
2. **Viability of Microbial Cells**:
   The concern about microbial cell viability is noted; however, this is not a critical factor for our analyses. The primary goal of the preparation was to retain intact cells, soil particles, and INPs bound to these components. Viability is not required neither for the INP analyses nor for the community composition analysis performed in this study.
3. **Mortaring**:
   Mortaring was conducted as a standard soil preparation step prior to sieving. This process ensured the breakdown of soil aggregates formed during freeze-drying, enabling effective sieving. We do not expect mortaring to cause significant degradation or loss of INPs within the samples.
4. **Sieving**:
   Sieving was employed to isolate particles smaller than 63 μm, which are more representative of those aerosolized directly from the soil. Analyzing bulk soil samples without sieving would not have accurately reflected the aerosolizable fraction of the soil.

We have incorporated the following justification into the manuscript:

*"We prepared soil samples for ice nucleation analysis as previously described, with slight modifications (Conen and Yakutin, 2018). The soil samples were placed in a small petri dish and freeze dried overnight (Edwards Micro Modulyo Freeze Dryer). Freeze-drying was chosen instead of air-drying to minimize the potential for microbial activity during the drying process and preserving the original composition of INPs, as prolonged air-drying could induce microbial activity, potentially altering INP concentrations due to production or degradation. The freeze-dried samples were kept in a desiccator to prevent rehydration and subsequently comminuted in a mortar by hand. Mortaring was performed to break down aggregates formed during freeze-drying and ensured effective sieving. Samples were dry sieved with a 125 μm and 63 μm sieve for two minutes using a vibratory sieve shaker (Analysette 3 PRO, Fritsch). The <63 μm fraction was collected for analysis, as this size range represents particles most likely to aerosolize (Fröhlich-Nowoisky et al., 2016). Hundred mg of dry <63 μm soil particles was weighed into an Eppendorf tube. For many samples, there was less than 0.1 g soil after sieving. Instead, all the sieved soil was added to the Eppendorf tube and the weight was noted. 1 mL filtered Milli-Q (0.22 μm PES) was added to the Eppendorf tube, then vortexed for two minutes and afterwards allowed to settle for 10 minutes. 0.5 mL was withdrawn from the top of the suspension and added to a falcon tube with 9.5 mL of filtered Milli-Q (0.22 μm) creating a 1:20 dilution. "*

Fig 3: This is a well-constructed summary figure; however, could the authors also include the other figures referenced on line 237 (Creamean et al., 2020; Schnell and Vali, 1976)? Incorporating these would provide a more comprehensive overview.

We thank the reviewer for acknowledging the value of Fig. 3 and for suggesting the inclusion of additional figures from Creamean et al. (2020) and Schnell and Vali (1976). Unfortunately, the data from Creamean et al. (2020) are not publacly accessible via the provided data availability statement https://stack.iop.org/ERL/15/084022/mmedia. Thus, we could not add them to the summary figure.

Regarding Schnell and Vali (1976), we note that their study focused primarily on leaf litter as a source of ice nucleating particles (INPs) rather than Arctic soil. While their findings are insightful in broader contexts, they are not directly relevant to our study of Arctic soil INPs. Therefore, to maintain the figure's focus on Arctic-specific sources, we have opted not to include this reference.

Line 242: Since the authors note that the Tobo et al. study focused on glacial outwash sediment, it would be helpful to specify that the Barry et al. study pertains to permafrost.

Thank you for this comment, which has been implemented in the new version of the manuscript.

Lines 243-244: How do these TC values compare to others in the literature for similar

soils?

We thank the reviewer for this question. The TC values in our samples were relatively low, with 9 out of 11 samples containing <5% w/w TC. We have compared these values to the literature and observed that they are lower than those reported by Conen and Yakutin (2018) for soils from Central Yakutia. However, our TC values are comparable to the glacial outwash sediments studied by Tobo et al. (2019). This comparison has been included in the revised manuscript to provide the necessary context.

*"A possible explanation for the lower concentration could be the rather low carbon content of the soils measured in this study with 9 out of 11 samples <5 % w/w TC, which is less than what Conen and Yakutin (2018) found in soils from Central Yakutia, but similar to the glacial outwash sediment that Tobo et al. (2019) investigated (Conen and Yakutin, 2018; Tobo et al., 2019)."*

Lines 245-246: More biomass and soil carbon content than what?

Thank you for this comment, this has now been clarified in the manuscript

Lines 251-253: Other factors contributing to the large variations may stem from the sample preparation methods. Conen et al. used sieving but did not employ mortaring, while Tobo et al. utilized neither technique. While differences in microbial community composition or soil properties could influence the results, the impact of the varying sample preparation methods should not be overlooked.

We refer to the reviewer's earlier comment (Lines 137-1) and our response.

Lines 254-255: While Santl-Temkiv et al. provides a valuable review on aerobiology, the authors should include other relevant papers as mentioned in previous comments.

Thank you for this comment we have now added Kanji et al. (2017) and Huang et al. (2021) to provide a broader overview.

Fig 3: The spectra from Conen et al. are somewhat difficult to distinguish. I recommend using a different color for clarity. Additionally, it would be beneficial to assign different colors and/or markers to the spectra from the 2011 and 2018 studies for better differentiation.

Thank you for this comment. The spectra have now been updated with slightly different grey colors for Conen et al 2011 and 2018, respectively, together with differing symbols.

Lines 271-277:
 This text would be better placed in the methods section.

Thank you for this comment. We have now specified the statistical approach in the methods section and shortened the text lines 271-277

*"To further characterize INPs within the Arctic soil, we used filtration analysis as different microorganisms produce INpro of different molecular sizes, which can either be firmly bound to the cells or easily removed resulting in soluble proteins (O'sullivan et al., 2015; Santl-Temkiv et al., 2022). A similar approach has previously been used to study the origin of INP in environmental samples (Conen and Yakutin, 2018; Fröhlich-Nowoisky et al., 2015). A Kruskal-Wallis test, indicated significant differences between the filtration treatments (p-value = 0.0001) (Fig. 4). A Wilcoxon rank sum test showed a significant difference between the bulk sample and the 300-100 kDa fraction (p = 0.0046). Subsequently, we analyzed the samples from individual locations to identify specific patterns. "*

Figs 4 (and 7): Technically, the sample should not be labeled as "bulk" as indicated on the x-axis. The authors should refer to this as "≤ 63 µm" instead.

Thank you for your comment. We have updated Figure 4 and supplementary Figure 1 to reflect that the "bulk" category refers to particles < 63 µm. However, we have retained Figure 7 as originally presented, as the water samples were not pretreated in this case.

Line 287: This analysis focuses on INP size and inferred composition, rather than direct composition. Additionally, it would be helpful to mention whether other studies, such as Barry et al. (2023a, b), observe significant variations in the INPs present in soil. The Barry et al. studies investigated concentrations and the effects of heat and peroxide treatments on composition (2023a and b), along with size filtering (2023b only).

Thank you for this comment. We have now added the word "inferred in front of the word "composition". Additionally, we discuss the results from Barry et al. 2023 a & b:

*Our findings somewhat align with Barry et al. (2023b), who demonstrated that the majority of INPs in soil were larger than 0.2 µm and primarily of biological origin. Furthermore, their heat treatment experiments revealed that INPs in permafrost soil are predominantly heat-labile, further supporting their biological nature. Interestingly, Barry et al. (2023b) reported a limited presence of soluble INPs smaller than 0.2 µm in permafrost soil, suggesting a scarcity of such low-molecular-weight biological INPs in their study system. This contrasts with the observed presence of INPs spanning a large range of sizes in our samples, including <100 kDa and aggregates >1000 kDa (Fig. 4). Such discrepancies might reflect differences in the environmental conditions, microbial communities, or soil composition between their study sites and ours. Additionally, Barry et al. (2023a) highlighted that INP concentrations in permafrost soil are influenced by particle size and composition, observing that larger particles (>10 µm) were significant contributors to INA in younger permafrost samples. While their findings pertain to larger particle fractions, our study emphasizes the role of smaller clay-bound particles (<5 µm) in transporting INPs into the atmosphere. This difference may reflect distinct mechanisms of INP atmospheric rentention time: larger particles contribute primarily to undisturbed permafrost soils and while quickly settle after aerosolization, while smaller,*

*clay-bound INPs being more prone to a longer atmospheric residence time after aerosolization.*

Line 304-205: The statement, "The gradual loss of INA during filtration at the different locations suggests a mixture of different-sized INPs, predominantly originating from fungi," needs clarification. Where is this information presented in the manuscript, or what other evidence supports this claim?

We thank the reviewer for this comment and agree that further clarification is necessary. In the revised manuscript, we have explicitly referenced Figure 4, which shows the data supporting this conclusion. Furthermore, we expanded the discussion to better explain the basis of this interpretation:

*The gradual loss of INA during filtration at the different locations suggests a mixture of different-sized INPs, predominantly originating from fungi (Fig. 4). This interpretation is supported by the fact that fungal INPs are known to span a wide range, including small <100 kDa (e.g., 5 kDa), and medium-sized molecules (100-300 kDa and 300-1000 kDa), and can bind to clay particles, resulting in INPs >1000 kDa and >0.2 μm (Kunert et al., 2019; Schwidetzky et al., 2023; O'sullivan et al., 2015; Conen and Yakutin, 2018). The size distribution, combined with the observed solubility and INA, aligns with the characteristics of fungal INPs, further supporting their major contribution to the INP pool in these soils.*

Lines 308-318: This is a nice summary, but would fit better in the conclusions section.

We appreciate the reviewers suggestion to move the summary (Lines 308–318) to the conclusions section. However, we believe this section is essential within the Results and Discussion because it integrates our findings with their broader implications, linking them directly to existing literature. The interpretation of fungal INPs as contributors to the observed ice-nucleating activity in Arctic soils builds on the presented evidence, such as size distribution, solubility, and INA characteristics. This placement allows us to connect our observations - such as the role of clay-bound particles and fungal INPs - to atmospheric processes and contextualize these results against previous studies (e.g., O'Sullivan et al., 2016; Kanji et al., 2017; Tobo et al., 2019). Moving this discussion to the conclusions risks detaching these connections from the results, which could dilute the integration of evidence and interpretation.

Line 316: Regarding the "upward fluxes," was the surface marshy or dry? Positive fluxes from the surface would depend on the surface aridity. This is an example of how describing the landscape of the sampling locations would be beneficial. Additionally, on line 390, wind erosion is mentioned; however, this also depends on surface aridity, which may not be realistic if the sampling locations were marshy.

Thank you for pointing out the relevance of surface conditions, such as aridity, to upward fluxes. While surface aridity is indeed a critical factor in aerosolization processes, we argue that the period of sampling in the Arctic is equally important when discussing

upward flux potential. In the Arctic, environmental conditions, including soil moisture and aridity, can vary substantially over the course of the melt season. Even if the surface appeared marshy on the specific day of sampling, it does not necessarily reflect the overall potential for aerosolization throughout the different seasons. As the seasons progresses, the soil surface in the Arctic often transitions from wet to increasingly dry due to diminishing snow and ice, coupled with higher evaporation rates and minimal precipitation. These drying phases are key drivers of upward fluxes, as drier surfaces are more prone to aerosolization under wind or disturbance.

As previously mentioned we have now included a section describing the sampling area, with reference to Riis et al. (2023), to provide further context.

Lines 337-344: The authors conclude that the abundances of known INP-producing species are very low for both 16S and ITS, with only sequences affiliated with Acremonium (at one location) and Mortierella (at most locations) present in their dataset. They suggest that the observed taxa might be INP producers that have not yet been recognized as such. However, could the INPs be derived from other organic materials, rather than exclusively from cellular or proteinaceous sources?

Thank you for the thoughtful observation. While we suggest that the observed taxa might include INP producers that are not yet recognized, we also acknowledge that INPs in the environment are not exclusively derived from cellular or proteinaceous sources. Organic materials such as polysaccharides, humic substances, and other macromolecules are known to exhibit ice-nucleating activity under certain conditions.
As discussed in a previous response, carbohydrates like cellulose and lignine have been shown to nucleate ice (Bogler and Borduas-Dedekind, 2020; Hiranuma et al., 2015); however, their nucleation activity generally occurs at significantly lower temperatures than what we observed in our study. The onset freezing temperatures and the freezeing profiles in our samples are indicative of proteinaceous INPs.

Lines 417-445: The authors should summarize and directly compare their findings to those of Barry et al. (2023a), as their INP results are derived from freshwater thermokarst lakes in the Arctic and are likely the most relevant for comparison with the Arctic stream water analyzed in this study. Barry et al. also included comparisons with locally sampled permafrost and active layer soils, while the other studies cited are focused on temperate regions.

Thank you this suggestion. In the revised manuscript, we have expanded the discussion in Section 3.4 to provide a direct comparison with Barry et al., as their work on Arctic thermokarst lakes is indeed the most relevant reference for our Arctic stream water data. This expanded discussion should provide a more robust comparison of our findings to Barry et al. (2023a) and other relevant studies, addressing the reviewer's concern:

*"In addition to characterizing soil INPs and their potential sources, we investigated the linkages between soil-freshwater INPs. The freezing onset was > -10°C for all twelve water sampling locations (Supplementary Fig. 7). The highest onset temperature was found in*

*Aucella (-5.9°C) and lowest in West 4 (-9.1°C). The high freezing temperatures indicate that the INPs are of biological origin (Kanji et al., 2017). The ice nucleation site density per volume of freshwater ($N_V$) as a function of temperature is shown in Fig. 6. The $INP_{-10}$ concentration measured in our study (average: 1005 $mL^{-1}$; range 24-4,880 $mL^{-1}$) (Table 1) are significantly lower than those reported by Barry et al. (2023a) for Arctic thermokarst lakes (average: 34300 $mL^{-1}$; range 1360-242,000 $mL^{-1}$). One potential explanation for this difference is the significantly higher soil INP concentrations reported in Barry et al.'s study compared to our measured soil INP concentrations. If soil is a major source of INPs to freshwater systems, as suggested by both studies, then lower soil INP concentrations in our sampling locations may directly contribute to the lower INP concentrations observed in Arctic streams relative to thermokarst lakes. "*

Lines 425-427: Huang et al. (2021) discuss how local Arctic sources can be rich in INPs, yet the concentration of aerosol INPs remains low. Given this context, is this finding truly surprising? Additionally, on lines 527-528, the authors state that "In streams, INP concentrations defied conventional expectations, exhibiting elevated concentrations contrary to the typical decrease towards polar regions." Is this assertion accurate?

Thank you for your valuable feedback. We agree with your assessment, and we have revised the manuscript to clarify these points. In response to your first comment, we have revised the discussion section (Lines 425-427) as follows:

"The high concentrations of $INP_{-10}$ in Arctic streams align with findings that several Arctic environments are rich in INPs, emphasizing their potential contribution as a regional source. However, despite these abundant local sources, aerosol INP concentrations in the Arctic atmosphere remain relatively low, possibly due to transport and deposition dynamics (Huang et al., 2021)."

Regarding your second comment on the assertion about INP concentrations in streams, we have updated the conclusion section to state:

*"In streams, INP concentrations were similar to those observed in temperate region rivers, such as the Mississippi and Gwaun Rivers. However, these concentrations were lower than those reported for other Arctic freshwater systems like thermokarst lakes."*

Lines 434-445: If these are all possible explanations, why would they apply specifically to the stream samples and not to the soil? This suggests that the INP populations in the two environments are not the same.

Thank you for this question. These explanations refer to the insoluble fraction of the INP population in the streams and hence not the soluble part. When looking only at the soluble part of INPs, which is likely the ones being transported from the soil to the streams it seems that the filtration experiment is in line with each other for both soil and stream INP as stated in lines: 445-449:

*Statistical analysis using the Kruskal-Wallis test showed a significant difference among the treatments (p-value = 1.721·10$^{-8}$). Post hoc Wilcoxon rank sum tests revealed a*

*significant change from bulk to the 300-100 kDa category (p = 0.0021). This was also observed in soil samples (Fig. 4), indicating that similar INPs are present in soil and streams which further imply the possible transfer of INPs from soil into the streams.*

Lines 487-489: Missing some key references here that looked at INPs in snowmelt, such as Brennan et al. (2020), Creamean et al. (2019), Stopelli et al. (2015, 2017). It would be useful to compare values to more than just Christner et al., (2008) and Santl-Temkiv et al. (2018).

Thank you for this comment, we have updated the text accordingly to show more examples of INP concentrations in snowmelt

*"Precipitation is unlikely to be a significant source of INPs in the studied streams, as INP concentrations in snow are typically much lower than those measured in the streams. Previous studies report $INP_{-10}$ concentrations in snow ranging from as low as $1.2 \cdot 10^{-2}$ $INP_{-10}$ $mL^{-1}$. to approximately $8 \cdot 10^{1}$ $INP_{-10}$ $mL^{-1}$, which are significantly lower than the values observed in stream water (Christner et al., 2008; Santl-Temkiv et al., 2019; Creamean et al., 2019; Brennan et al., 2020). "*

Lines 536-537: The statement, "...future research should focus on deciphering the contributions from various sources such as soil, runoff, and marine emissions to fully elucidate their roles in cloud formation and climate processes," should acknowledge the work of Barry et al. (2023a), who investigated a wide range of potential sources, including those mentioned, and linked their findings to INP data collected upwind and downwind of thermokarst lakes. They should receive appropriate credit for their contributions in this context.

Thank you for the comment, we have now implemented the acknowledgement in the conclusion:

*"While the direct quantification of aerosolization was beyond the scope of this study, future research should focus on deciphering the contributions from various sources, such as active layer soil, runoff, and marine emissions, combining the approaches used in this study with those employed in studies like Barry et al. (2023a, 2023b), to fully elucidate their roles in cloud formation and climate processes."*

Supplemental Figs 3 and 4: These figures seem central to the main takeaways, why are they not shown in the main text?

We appreciate the reviewer's suggestion regarding the inclusion of Supplemental Figures 3 and 4, which display the phylum-level bacterial and fungal relative abundances. We agree that these figures provide valuable context for understanding the microbial community composition. However, we believe that they serve primarily as an overview and do not directly contribute to the central conclusions of our study. The main focus of our manuscript lies in linking specific microbial genera to INP activity, which is more effectively illustrated by Figure 5, showing the significant correlations between microbial taxa and INP concentrations.

Including supplementary Figures 3 and 4 in the main text could distract from the key findings and potentially overwhelm the reader with less central information. Therefore, we have chosen to retain these figures in the supplement, where they provide additional context without detracting from the main narrative. We hope this approach maintains the clarity and focus of our message.

References:
Alpert, P. A., Kilthau, W. P., O'Brien, R. E., Moffet, R. C., Gilles, M. K., Wang, B., Laskin, A., Aller,
J. Y., and Knopf, D. A.: Ice-nucleating agents in sea spray aerosol identified and quantified with a holistic multimodal freezing model, Science Advances, 8, eabq6842, https://doi.org/10.1126/sciadv.abq6842, 2022.
Barry, K. R., Hill, T. C. J., Nieto-Caballero, M., Douglas, T. A., Kreidenweis, S. M., DeMott, P.
J., and Creamean, J. M.: Active thermokarst regions contain rich sources of ice-nucleating particles, Atmospheric Chemistry and Physics, 23, 15783–15793, https://doi.org/10.5194/acp-23-15783-2023, 2023a.
Barry, K. R., Hill, T. C. J., Moore, K. A., Douglas, T. A., Kreidenweis, S. M., DeMott, P. J., and Creamean, J. M.: Persistence and Potential Atmospheric Ramifications of Ice-Nucleating Particles Released from Thawing Permafrost, Environ. Sci. Technol., 57, 3505–3515, https://doi.org/10.1021/acs.est.2c06530, 2023b.
Bigg, E. K.: Ice forming nuclei in the high Arctic, 1996.
Bigg, E. K. and Leck, C.: Cloud-active particles over the central Arctic Ocean, J. Geophys. Res., 106, 32155–32166, https://doi.org/10.1029/1999JD901152, 2001.
Brennan, K. P., David, R. O., and Borduas-Dedekind, N.: Spatial and temporal variability in the ice-nucleating ability of alpine snowmelt and extension to frozen cloud fraction, Atmospheric Chemistry and Physics, 20, 163–180, https://doi.org/10.5194/acp-20-163-2020, 2020.
Cartmill, J. W. and Yang Su, M.: Bubble size distribution under saltwater and freshwater breaking waves, Dynamics of Atmospheres and Oceans, 20, 25–31, https://doi.org/10.1016/0377-0265(93)90046-A, 1993.
Creamean, J. M., Mignani, C., Bukowiecki, N., and Conen, F.: Using freezing spectra characteristics to identify ice-nucleating particle populations during the winter in the Alps, Atmospheric Chemistry and Physics, 19, 8123–8140, https://doi.org/10.5194/acp-19-8123-2019, 2019.
Creamean, J. M., Barry, K., Hill, T. C. J., Hume, C., DeMott, P. J., Shupe, M. D., Dahlke, S., Willmes, S., Schmale, J., Beck, I., Hoppe, C. J. M., Fong, A., Chamberlain, E., Bowman, J., Scharien, R., and Persson, O.: Annual cycle observations of aerosols capable of ice formation in central Arctic clouds, Nat Commun, 13, 3537, https://doi.org/10.1038/s41467-022-31182-x, 2022.
Harrison, A. D., Lever, K., Sanchez-Marroquin, A., Holden, M. A., Whale, T. F., Tarn, M. D., McQuaid, J. B., and Murray, B. J.: The ice-nucleating ability of quartz immersed in water and its atmospheric importance compared to K-feldspar, Atmospheric Chemistry and Physics, 19, 11343–11361, https://doi.org/10.5194/acp-19-11343-2019, 2019.

Hartmann, M., Adachi, K., Eppers, O., Haas, C., Herber, A., Holzinger, R., Hünerbein, A., Jäkel, E., Jentzsch, C., Pinxteren, M., Wex, H., Willmes, S., and Stratmann, F.: Wintertime Airborne Measurements of Ice Nucleating Particles in the High Arctic: A Hint to a Marine, Biogenic Source for Ice Nucleating Particles, Geophys. Res. Lett., 47, https://doi.org/10.1029/2020GL087770, 2020.

Hartmann, M., Gong, X., Kecorius, S., Van Pinxteren, M., Vogl, T., Welti, A., Wex, H., Zeppenfeld, S., Herrmann, H., Wiedensohler, A., and Stratmann, F.: Terrestrial or marine – indications towards the origin of ice-nucleating particles during melt season in the European Arctic up to 83.7° N, Atmos. Chem. Phys., 21, 11613–11636, https://doi.org/10.5194/acp-21-11613-2021, 2021.

Ickes, L., Porter, G. C. E., Wagner, R., Adams, M. P., Bierbauer, S., Bertram, A. K., Bilde, M., Christiansen, S., Ekman, A. M. L., Gorokhova, E., Höhler, K., Kiselev, A. A., Leck, C., Möhler, O., Murray, B. J., Schiebel, T., Ullrich, R., and Salter, M.: Arctic marine ice nucleating aerosol: a laboratory study of microlayer samples and algal cultures, Aerosols/Laboratory Studies/Troposphere/Physics (physical properties and processes), https://doi.org/10.5194/acp-2020-246, 2020a.

Ickes, L., Porter, G. C. E., Wagner, R., Adams, M. P., Bierbauer, S., Bertram, A. K., Bilde, M., Christiansen, S., Ekman, A. M. L., Gorokhova, E., Höhler, K., Kiselev, A. A., Leck, C., Möhler, O., Murray, B. J., Schiebel, T., Ullrich, R., and Salter, M. E.: The ice-nucleating activity of Arctic sea surface microlayer samples and marine algal cultures, Atmos. Chem. Phys., 20, 11089–11117, https://doi.org/10.5194/acp-20-11089-2020, 2020b.

Jayaweera, K. and Flanagan, P.: Investigations on biogenic ice nuclei in the Arctic atmosphere, Geophysical Research Letters, 9, 94–97, https://doi.org/10.1029/GL009i001p00094, 1982.

Kanji, Z. A., Ladino, L. A., Wex, H., Boose, Y., Burkert-Kohn, M., Cziczo, D. J., and Krämer, M.: Overview of Ice Nucleating Particles, https://doi.org/10.1175/AMSMONOGRAPHS-D-16-0006.1, 2017.

Murray, B. J., O'Sullivan, D., Atkinson, J. D., and Webb, M. E.: Ice nucleation by particles immersed in supercooled cloud droplets, Chem. Soc. Rev., 41, 6519–6554, https://doi.org/10.1039/C2CS35200A, 2012.

Porter, G. C. E., Adams, M. P., Brooks, I. M., Ickes, L., Karlsson, L., Leck, C., Salter, M. E., Schmale, J., Siegel, K., Sikora, S. N. F., Tarn, M. D., Vüllers, J., Wernli, H., Zieger, P., Zinke, J., and Murray, B. J.: Highly Active Ice-Nucleating Particles at the Summer North Pole, Journal of Geophysical Research: Atmospheres, 127, e2021JD036059, https://doi.org/10.1029/2021JD036059, 2022.

Schnell, R. C. and Vali, G.: Biogenic Ice Nuclei: Part I. Terrestrial and Marine Sources, J. Atmos. Sci., 33, 1554–1564, 1976.

Stopelli, E., Conen, F., Morris, C. E., Herrmann, E., Bukowiecki, N., and Alewell, C.: Ice nucleation active particles are efficiently removed by precipitating clouds, Sci Rep, 5, 16433, https://doi.org/10.1038/srep16433, 2015.

Stopelli, E., Conen, F., Guilbaud, C., Zopfi, J., Alewell, C., and Morris, C. E.: Ice nucleators, bacterial cells and Pseudomonas syringae; in precipitation at Jungfraujoch, Biogeosciences, 14, 1189–1196, https://doi.org/10.5194/bg-14-1189-2017, 2017.

Tobo, Y.: An improved approach for measuring immersion freezing in large droplets over a wide temperature range, Sci Rep, 6, 32930, https://doi.org/10.1038/srep32930, 2016.

Vali, G., Christensen, M., Fresh, R. W., Galyan, E. L., Maki, L. R., and Schnell, R. C.: Biogenic
Ice Nuclei. Part II: Bacterial Sources, Journal of the Atmospheric Sciences, 33, 1565–1570, https://doi.org/10.1175/1520-0469(1976)033<1565:BINPIB>2.0.CO;2, 1976.

Zinke, J., Nilsson, E. D., Zieger, P., and Salter, M. E.: The Effect of Seawater Salinity and Seawater Temperature on Sea Salt Aerosol Production, JGR Atmospheres, 127, https://doi.org/10.1029/2021JD036005, 2022.

Additional references:

Bogler, S. and Borduas-Dedekind, N.: Lignin's ability to nucleate ice via immersion freezing and its stability towards physicochemical treatments and atmospheric processing, Atmospheric Chemistry and Physics, 20, 14509-14522, 10.5194/acp-20-14509-2020, 2020.

Brennan, K. P., David, R. O., and Borduas-Dedekind, N.: Spatial and temporal variability in the ice-nucleating ability of alpine snowmelt and extension to frozen cloud fraction, Atmospheric Chemistry and Physics, 20, 163-180, 10.5194/acp-20-163-2020, 2020.

Christiansen, H. H., Sigsgaard, C., Humlum, O., Rasch, M., and Hansen, B. U.: Permafrost and Periglacial Geomorphology at Zackenberg, in: High-Arctic Ecosystem Dynamics in a Changing Climate, Advances in Ecological Research, 151-174, 10.1016/s0065-2504(07)00007-4, 2008.

Christner, B. C., Morris, C. E., Foreman, C. M., Cai, R., and Sands, D. C.: Ubiquity of biological ice nucleators in snowfall, Science, 319, 1214, 10.1126/science.1149757, 2008.

Conen, F. and Yakutin, M. V.: Soils rich in biological ice-nucleating particles abound in ice-nucleating macromolecules likely produced by fungi, Biogeosciences, 15, 4381-4385, 10.5194/bg-15-4381-2018, 2018.

Cornwell, G. C., McCluskey, C. S., Hill, T. C. J., Levin, E. T., Rothfuss, N. E., Tai, S.-L., Petters, M. D., DeMott, P. J., Kreidenweis, S., Prather, K. A., and Burrows, S. M.: Bioaerosols are the dominant source of warm-temperature immersion-mode INPs and drive uncertainties in INP predictability, Science Advances, 9, eadg3715, doi:10.1126/sciadv.adg3715, 2023.

Creamean, J. M., Mignani, C., Bukowiecki, N., and Conen, F.: Using freezing spectra characteristics to identify ice-nucleating particle populations during the winter in the Alps, Atmospheric Chemistry and Physics, 19, 8123-8140, 10.5194/acp-19-8123-2019, 2019.

Daily, M. I., Tarn, M. D., Whale, T. F., and Murray, B. J.: An evaluation of the heat test for the ice-nucleating ability of minerals and biological material, Atmospheric Measurement Techniques, 15, 2635-2665, 10.5194/amt-15-2635-2022, 2022.

Docherty, C. L., Dugdale, S. J., Milner, A. M., Abermann, J., Lund, M., and Hannah, D. M.: Arctic river temperature dynamics in a changing climate, River Research and Applications, 35, 1212-1227, 10.1002/rra.3537, 2019.

Fröhlich-Nowoisky, J., Hill, T. C. J., Pummer, B. G., Yordanova, P., Franc, G. D., and Pöschl, U.: Ice nucleation activity in the widespread soil fungus Mortierella alpina, Biogeosciences, 12, 1057-1071, 10.5194/bg-12-1057-2015, 2015.

Fröhlich-Nowoisky, J., Kampf, C. J., Weber, B., Huffman, J. A., Pöhlker, C., Andreae, M. O., Lang-Yona, N., Burrows, S. M., Gunthe, S. S., Elbert, W., Su, H., Hoor, P., Thines, E., Hoffmann, T., Després, V. R., and Pöschl, U.: Bioaerosols in the Earth system: Climate, health, and ecosystem interactions, Atmospheric Research, 182, 346-376, 10.1016/j.atmosres.2016.07.018, 2016.

Hartmann, S., Ling, M., Dreyer, L. S. A., Zipori, A., Finster, K., Grawe, S., Jensen, L. Z., Borck, S., Reicher, N., Drace, T., Niedermeier, D., Jones, N. C., Hoffmann, S. V., Wex, H., Rudich, Y., Boesen, T., and Santl-Temkiv, T.: Structure and Protein-Protein Interactions of Ice Nucleation Proteins Drive Their Activity, Front Microbiol, 13, 872306, 10.3389/fmicb.2022.872306, 2022.

Hasholt, B. and Hagedorn, B.: Hydrology and Geochemistry of River-Borne Material in a High Arctic Drainage System, Zackenberg, Northeast Greenland, Arctic, Antarctic, and Alpine Research, 32, 84-94, 10.2307/1552413, 2000.

Hiranuma, N., Möhler, O., Yamashita, K., Tajiri, T., Saito, A., Kiselev, A., Hoffmann, N., Hoose, C., Jantsch, E., Koop, T., and Murakami, M.: Ice nucleation by cellulose and its potential contribution to ice formation in clouds, Nat Geosci, 8, 273-277, 10.1038/ngeo2374, 2015.

Hollesen, J., Elberling, B., and Jansson, P. E.: Future active layer dynamics and carbon dioxide production from thawing permafrost layers in Northeast Greenland, Global Change Biol, 17, 911-926, 10.1111/j.1365-2486.2010.02256.x, 2011.

Kanji, Z. A., Ladino, L. A., Wex, H., Boose, Y., Burkert-Kohn, M., Cziczo, D. J., and Krämer, M.: Overview of Ice Nucleating Particles, Meteorological Monographs, 58, 1.1-1.33, https://doi.org/10.1175/AMSMONOGRAPHS-D-16-0006.1, 2017.

Kinney, N. L. H., Hepburn, C. A., Gibson, M. I., Ballesteros, D., and Whale, T. F.: High interspecific variability in ice nucleation activity suggests pollen ice nucleators are incidental, Biogeosciences, 21, 3201-3214, 10.5194/bg-21-3201-2024, 2024.

Knackstedt, K. A., Moffett, B. F., Hartmann, S., Wex, H., Hill, T. C. J., Glasgo, E. D., Reitz, L. A., Augustin-Bauditz, S., Beall, B. F. N., Bullerjahn, G. S., Fröhlich-Nowoisky, J., Grawe, S., Lubitz, J., Stratmann, F., and McKay, R. M. L.: Terrestrial Origin for Abundant Riverine Nanoscale Ice-Nucleating Particles, Environmental Science & Technology, 52, 12358-12367, 10.1021/acs.est.8b03881, 2018.

Kunert, A. T., Pöhlker, M. L., Tang, K., Krevert, C. S., Wieder, C., Speth, K. R., Hanson, L. E., Morris, C. E., Schmale Iii, D. G., Pöschl, U., and Fröhlich-Nowoisky, J.: Macromolecular fungal ice nuclei in Fusarium: effects of physical and chemical processing, Biogeosciences, 16, 4647-4659, 10.5194/bg-16-4647-2019, 2019.

Larsen, J. A., Conen, F., and Alewell, C.: Export of ice nucleating particles from a watershed, R Soc Open Sci, 4, 170213, 10.1098/rsos.170213, 2017.

Mankoff, K. D., Noël, B., Fettweis, X., Ahlstrøm, A. P., Colgan, W., Kondo, K., Langley, K., Sugiyama, S., van As, D., and Fausto, R. S.: Greenland liquid water discharge from 1958 through 2019, Earth System Science Data, 12, 2811-2841, 10.5194/essd-12-2811-2020, 2020.

Murray, B. J., O'Sullivan, D., Atkinson, J. D., and Webb, M. E.: Ice nucleation by particles immersed in supercooled cloud droplets, Chem Soc Rev, 41, 6519-6554, 10.1039/c2cs35200a, 2012.

O'Sullivan, D., Murray, B. J., Ross, J. F., Whale, T. F., Price, H. C., Atkinson, J. D., Umo, N. S., and Webb, M. E.: The relevance of nanoscale biological fragments for ice nucleation in clouds, Sci Rep, 5, 8082, 10.1038/srep08082, 2015.

Polen, M., Brubaker, T., Somers, J., and Sullivan, R. C.: Cleaning up our water: reducing interferences from nonhomogeneous freezing of "pure" water in droplet freezing assays of ice-nucleating particles, Atmos. Meas. Tech., 11, 5315-5334, 10.5194/amt-11-5315-2018, 2018.

Santl-Temkiv, T., Amato, P., Casamayor, E. O., Lee, P. K. H., and Pointing, S. B.: Microbial ecology of the atmosphere, FEMS Microbiol Rev, 10.1093/femsre/fuac009, 2022.

Santl-Temkiv, T., Lange, R., Beddows, D., Rauter, U., Pilgaard, S., Dall'Osto, M., Gunde-Cimerman, N., Massling, A., and Wex, H.: Biogenic Sources of Ice Nucleating Particles at the High Arctic Site Villum Research Station, Environ Sci Technol, 53, 10580-10590, 10.1021/acs.est.9b00991, 2019.

Schwidetzky, R., de Almeida Ribeiro, I., Bothen, N., Backes, A. T., DeVries, A. L., Bonn, M., Frohlich-Nowoisky, J., Molinero, V., and Meister, K.: Functional aggregation of cell-free proteins enables fungal ice nucleation, Proc Natl Acad Sci U S A, 120, e2303243120, 10.1073/pnas.2303243120, 2023.

Tobo, Y., Adachi, K., DeMott, P. J., Hill, T. C. J., Hamilton, D. S., Mahowald, N. M., Nagatsuka, N., Ohata, S., Uetake, J., Kondo, Y., and Koike, M.: Glacially sourced dust as a potentially significant source of ice nucleating particles, Nat Geosci, 12, 253-258, 10.1038/s41561-019-0314-x, 2019.

---

## Author Comment (AC2)

The manuscript presents a comprehensive study of INP concentration and size in soil and streams around Zackenberg, eastern Greenland. Further investigations point out members of the soil microbial community potentially having formed these INPs. Overall, Jensen et al. have managed to analyse and interpret their diverse results in a coherent, logically consistent way. I agree with Anonymous Referee #1 that the manuscript is interesting and should be published. In addition to their detailed review, I have two thoughts the authors may consider when revising their manuscript.

1.) When looking at Figures 1 and 2 placed next to each other, I get the impression that shorter streams on steep terrain tend to carry lower concentrations of INPs (e.g.: West 4, West 5) as compared with longer streams (e.g.: West 1, West 3). Longer streams also tend to have sections on less steep terrain, where drainage water likely percolates slowly through soil and, therefore, has time to accumulate INPs. For proper quantitative analysis one would have to estimate the time snowmelt or rain water has spent in soil before entering a stream. Of course, such an estimate is well beyond the scope of this already comprehensive study. More easily, the length or average slope of streams could be derived from Figure 1 and used to put this idea to the test.

We sincerely appreciate your insightful comment, which offered an exciting new perspective on our data. Your suggestion that shorter streams in steep terrain may have lower INP concentrations due to limited time for water to percolate through soils, while longer streams in less steep terrain may accumulate more INPs, aligns well with our interest in the processes driving INP transport. Inspired by your idea, we examined potential correlations between stream $INP_{-10}$ concentrations, catchment area size, and slope. However, our analysis did not find significant relationships, suggesting that other factors such as hydrological dynamics, streamflow dilution, or variability in soil INP sources may play larger roles in modulating stream INP concentrations.

Nonetheless, your comment raises an exciting direction for future work to further investigate how terrain and hydrology interact to influence INP transport. This perspective significantly enriched our discussion and has been incorporated into the manuscript. Thank you for bringing this to our attention!

*"If soil INPs are a major source of INPs in streams, we would expect to observe a positive correlation between soil and stream INP concentrations. Furthermore, in areas with steep terrain, less time is available for water to percolate through soil and accumulate INPs before entering the stream. Larger catchment areas, typically associated with longer streams, might also increase the potential for INP accumulation due to extended flow paths and surface interactions. Surprisingly, we did not observe significant correlations between $INP_{-10}$ concentrations in streams and catchment area size, or the slope of the terrain. Only a weak, non-significant correlation between soil and stream $INP_{-10}$ concentrations was found (R = 0.23, p > 0.05). These findings suggest that while soil may contribute INPs to streams, other factors such as local hydrology, and dilution effects likely obscure a clear relationship. "*

2.) Although there is a general association of stream INP concentration with soil INP concentration, there is a striking difference in INP spectra between soil (Figure 2) and stream water (Figure 6). Whereas the former spectra are shallow above -10°C and mostly extend to above -5°C, the latter spectra show a steep decrease above -10°C and none extends to above -5°C. In other words, the most efficient INPs do not seem to be transferred from soil to stream water. One explanation could be that such INPs are too large to pass with draining water through the soil matrix. Another, that they lose their efficiency quickly after having been produced, more quickly than they are transferred to the stream.

We thank the reviewer for this insightful question.

The observation of a contrast between the INP spectra of soil and stream water, particularly in the efficiency of INPs active at higher temperatures, suggests two things: One plausible explanation is the dilution effect. The INPs in soil are likely concentrated in specific, localized regions, while the transfer to stream water, through the drainage process, would involve significant dilution. This dilution would disproportionately affect the more scarce, highly active INPs that exhibit activity at higher temperatures (e.g., around -5°C and above). These INPs are less abundant than those active at lower temperatures (e.g., below -10°C) and are likely diluted to the point where their concentrations are too low to be detected without prior concentration. As a result, the direct measurement of stream samples without concentrating them may miss these more active, but rarer, INPs, thereby giving the impression that they are not effectively transferred from soil to stream water.
Another factor to consider is the size distribution of INPs. Larger INPs, which are often more efficient nucleators, may be less mobile and more likely to be retained within the soil matrix. These larger particles may not pass through the soil as easily, especially if the pore spaces are small or the soil matrix is compacted. Therefore, the most efficient INPs, which tend to be larger, may be physically excluded from the stream water during the drainage process as the reviewer proposes.

Lines 483 and 493: Instead of "p > 0.05" I would prefer to the exact p-value (e.g., p = 0.08).

Thank you for your comment. This has been implemented in the new version.

---

## Author Comment (AC3)

Jensen et al. present a novel and interesting study of soils and streamwater from northeast Greenland and probe the biological composition of their samples using filtration and a number of DNA analysis techniques, finding correlations between bacterial and primarily fungal species with INP concentrations. The manuscript is well written and the work is generally very thorough, while the results add to our knowledge of INPs in the environment and open up new avenues of exploration. This manuscript is suitable for publication pending a handful of major questions and comments below.

Major comments:

1) Line 253 (and throughout the paper when referring to onset temperatures): "Our results showed higher onset temperatures (between -1.5 °C and -4.7 °C) compared to previous studies of Arctic soils (Fig. 3)" and the sentences thereafter – This could be a result of using larger droplet volumes (30 ul) in the droplet freezing assays compared to the literature, allowing the rarer particles to be detected (i.e. better sensitivity) rather than the soils being more active here than elsewhere. This needs to be discussed, at least as a caveat. Generally speaking, the use of onset temperatures or T50 values can only be compared in "like-for-like" experiments, and are not necessarily suitable for literature comparisons.

We agree with this concern regarding the comparability of onset freezing temperatures across studies using different droplet volumes or different input amount of soils in freezing assays. We have revised the manuscript to explicitly discuss the potential implications of methodological differences. Specifically, we added a section that acknowledges that larger droplet volumes are more sensitive to detecting rarer INPs. Many of the studies we compare our results to, used larger volumes and therefore their sensitivity is higher than in our assays, supporting the statement that INPs observed in our study indded show a higher onset temperature than other studies. In these revisions, we now emphasize both the methodological context and the unique INP activity observed in the Arctic soils we studied. By addressing these points, we believe the discussion now provides a balanced interpretation of the results while maintaining scientific rigor.

*"Our results showed higher onset temperatures (between -1.5 °C and -4.7 °C) compared to previous studies of Arctic soils (Fig. 3). Methodological differences, such as droplet volume used in freezing assays, must be considered when interpreting this trend. Studies using smaller volumes (e.g. 5 µL in Tobo et al., 2019), have a lower sensitivy and cannot be directly compared to our study. However, several studies used larger droplet volumes (e.g., 50 µL in Conen et al., 2018; Barry et al., 2023b, and 100 µL in Conen et al., 2012), which have a higher sensitivity than the micro-Pinguin asssay and a comparable potential to detect rare highly active INPs. Therefore, the higher freezeing onset that we observe does not seem to only be linked to methodological differences but reflects differences in the INP popoulations in these environments. INPs active at such high temperatures are generally proteinaceous (Santl-Temkiv et al., 2022) and are often associated with microbial sources, including bacteria and fungi (Barry et al., 2023b; Tobo et al., 2019; Conen et al., 2011). The presence of higher onset temperatures in this study may indicate differences in either the identity or in the activity of their microbial producers across Arctic terrestrial environments."*

2) Figure 2 and 6: Some of the data goes below -25oC, but it is not written or shown anywhere in the paper or Supporting Information what the freezing temperatures of the pure water controls were. In what range do the pure water droplets freeze and does it overlap with the sample data? It would be helpful to include the pure water data in the fraction frozen plots in the Supporting Information. If there is overlap between the control and sample data then

background-corrections may need to be applied, as per Vali 2019 (https://amt.copernicus.org/articles/12/1219/2019/).

Thank you for pointing this out. We have added the control data to the fraction frozen plots, as shown in Supplementary Figures 1 and 7. These controls demonstrate onset freezing temperatures consistently below -15°C. Therefore, background subtraction is not required in the temperature range of primary interest, specifically between 0°C and -15°C, as there is no overlap between the control data and the sample data.

3)    The streams are all defined as freshwater, but was there any formal analysis of their salinity, even if low? Did salinity levels vary at all across the streams (some samples appear to be from near the coastline) and could this be reflected in the INP spectra, e.g. do the INP concentrations decrease with salinity?

Thank you for your observation. We analyzed the salinity across the streams, measuring concentrations of $Na^+$ and $Cl^-$ ions to calculate NaCl levels and salinity. Salinity values ranged from 0.0012 ppt to 0.0184 ppt, confirming that all streams fall well within the freshwater range (below 0.5 ppt). Even for the stream with the highest salinity (Kærelv C, 0.0184 ppt), the salinity is extremely low, and any freezing point depression would be negligible.

4)    Supplementary figure 8 should ideally be in the main paper, particularly being that it is the equivalent of Figure 5 for the stream samples. Most of the factors shown in the figure are not discussed in the main paper but should also be mentioned.

Thank you for your comment. We appreciate the suggestion to move Supplementary Figure 8 to the main paper. The figure is primarily intended to support specific arguments made in the discussion, rather than to present critical results. Considering the fact that the correlations displayed in the figure do not reveal any significant novel findings, we decided to keep the figure in the SI to keep the manuscript focussed on the key findings.

5)    I was expecting more of a clear discussion about the links or their absence between the soil and water studies, for example what portion of microbial species were found in the soils that then also appeared in the water samples, and how did their relative amounts change, particularly for those that were ice nucleating. A weak positive correlation of INP concentration is mentioned in part 3.5, but there is not much discussion about the nature of the INPs between the two samples (unless I have missed it). This is a little lacking considering the abstract and introduction point to this link, e.g. "In addition, the transfer of bioINPs from soils into freshwater and marine systems has not been quantified. This study aimed at addressing these open questions…". Perhaps there is a reason why it may not be suitable to discuss, but it is not clear, and so if possible I would like to see at least some discussion of the potential links between the soils and streams, or otherwise make clear that this is not one of the points of the manuscript.

Thank you for your insightful comment. We agree that understanding the microbial links between soils and streams would be very important. Unfortunately, in this study, we could not conduct amplicon sequencing on the stream microbial community due to technical difficulties during the field campaign. This limits our ability to directly track specific microbial taxa or quantify their transfer from soils to streams. We have revised the manuscript to acknowledge this limitation and added a sentence in the discussion section emphasizing the need for future

studies that include parallel analyses of both soil and stream microbial communities. This would allow for a more detailed assessment of microbial transfer dynamics, particularly for ice-nucleating taxa.

*"To better understand the microbial transfer dynamics between soils and streams, future studies should include parallel analyses of both soil and stream microbial communities. This would enable a more detailed assessment of microbial transfer, particularly for ice-nucleating taxa, which is crucial for understanding the links between these two environments."*

6)   The plots should have error bars where possible, for example the INP plots.

Thank you for your suggestion. The analysis of INP data in this study does not involve binning the data into temperature intervals, as we rely on cumulative freezing spectra derived from individual droplets or wells. This approach provides precise freezing temperatures for each droplet, without aggregating the data into bins. Consequently, there is no statistical variation across temperature intervals to calculate error bars.
Furthermore, each sample is analyzed using 80 droplets, instead of analyzing smaller numbers of droplets in replicates, as has been recommended by Polen et al (2018) (Polen et al., 2018). This also means a high coverage of the freezing temperature distribution within the sample. This high number of droplets allows us to capture the variability of freezing events across the sample population with high confidence, minimizing the need for additional statistical representation such as error bars.

Minor comments:
1)   Line 304: "The gradual loss of INA during filtration at the different locations suggests a mixture of different-sized INPs, predominantly originating from fungi." – What would suggest that they cannot originate from bacteria?

Thank you for your comment. The statement that the INPs predominantly originate from fungi might at this point in the manuscript be a to strong statement. However, there are some evidence that they might be the predominant type compared to bacterial INpro:

**Retention of Activity After 0.2 μm Filtration:** In most samples, INPs retained activity after filtration through a 0.2 μm filter, which suggests that they are not membrane-bound. Bacterial INpro are typically associated with the bacterial outer membrane, as demonstrated in the literature (Santl-Temkiv et al., 2022). Retention of activity post-filtration is therefore inconsistent with a predominantly bacterial origin.
1.   **Size Distribution of Soluble INPs:** Soluble INPs in the majority of samples were found to be smaller than 0.2 μm while  >1000 kDa, consistent with fungal INpro aggregates or oligomers. For example, Fusarium acuminatum produces INpro aggregates up to ~700 kDa (Schwidetzky et al., 2023a). Others samples were more similar to INpro produced by Mortierella sp. and Fusarium sp. in the range of 300–100 kDa (Kunert et al., 2019; Schwidetzky et al., 2023). These size ranges align with the predominant INPs observed in our samples after filtration.
2.   **Binding to Clay Particles:** The observed decrease in activity in some samples (e.g., West 4 and Aucella) following filtration could be explained by fungal INPs binding to clay particles, which are abundant across all soil samples (Supplementary Fig. 2). Such

interactions are well-documented for fungal INPs (O'Sullivan et al., 2016), while bacterial INPs are not typically associated with clay particles in this manner.

Given this body of evidence, we have concluded that the gradual loss of INA during filtration is most likely due to a mixture of different-sized INPs predominantly originating from fungi and bacteria.

This has now been updated in the manuscript:

*"The gradual loss of INA during filtration at the different locations suggests a mixture of different-sized INPs, predominantly originating from fungi and bacteria (Fig. 4)."*

2) Was there any consideration of using heat treatments or peroxide treatments of the samples followed by reanalysis of the droplet freezing temperatures? These treatments have high uncertainties in that they do not necessarily "prove" the presence of biological (or entities produced by biological species), but can be a useful indication. On the other hand, DNA analysis allows direct detection of biological species including identification and even quantification, but appears to suffer from other issues, for example PCR would be used for the identification of specific known INP species (but could miss others), while sequencing informs on the identification of populations but not whether they are INPs or produce INpro. While not ideal, heat or peroxide treatments would at least allow an indication of the potential impact of INpro versus the mineral or clay particles that would presumably form the "background" signal.

We appreciate the reviewer's suggestion regarding heat treatments or peroxide treatments followed by reanalysis of droplet freezing temperatures. It is well-established in the literature that highly active INPs are predominantly of biological origin. Specifically, temperature treatments have been shown to significantly reduce the activity of INPs and close to 100% of INP-10 activity was lost after heat treatments (Daily et al., 2022; Barry et al., 2023b; Barry et al., 2023a). While heat treatments would confirm proteinaceous origin of the INPs that we observed, they would have not given specific insights into the microbial producers of the INPs. We therefore chose to perform filtration analysis as well as microbial community analysis attempting to more specificaly identifying the INP-producers.

3) Lines 420-429: While the soil results were compared to the literature in Figure 3, there is no such figure for the stream water results despite a description of several relevant datasets. While not essential, this would be easier to follow in a visual format rather than trying to compare numbers.

Thank you for the suggestion, which we agree with. However, most cited datasets are unfortunately not publically available and we have therefore decided not to include such a figure.

4) Could the fraction frozen and Nm/Nv data for the filtered samples be included in the Supplementary figures? Only the T50 values are discussed but this does not show whether there were any other influences on the INP populations, for example changes in the shape of the Nm curves upon filtering.

Thank you for your suggestion. We have now included the fraction frozen data for the filtered samples in Supplementary Figures 1 and 7.

5) What is the temperature uncertainty of the micro-PINGUIN technique?

The measurement accuracy of the micro-PINGUIN instrument is primarily influenced by the vertical temperature gradient within the well. A detailed breakdown of the uncertainty contributions can be found in (Wieber et al., 2024), Section 2.5 and Appendix A1–A7.
In summary, the largest contribution to the uncertainty arises from the vertical gradient in the well. This gradient was measured to be 0.20 °C at 0 °C, increasing by 0.015 °C for each degree below 0 °C. To account for this, temperature readings are corrected by half of the vertical gradient at each temperature point, ensuring that the surface temperature measured by the infrared camera is accurate.
We have included Table A1 from Wieber et al. (2024), which demonstrates that while the temperature correction increases with decreasing temperature, it never exceeds 1 °C. Hence, in the temperature range that we are interested in ie., 0 to -15 °C the uncertainty will be quite small and within the range of other ice nucleation setups uncertainty which is usually ranging between 0.5 to 1 °C (Lacher et al., 2024).

| Temperature | Correction | Uncertainty ($k = 2$) |
|---|---|---|
| 0 °C | −0.27 °C | ±0.59 °C |
| −5 °C | −0.38 °C | ±0.70 °C |
| −10 °C | −0.48 °C | ±0.81 °C |
| −15 °C | −0.59 °C | ±0.93 °C |
| −20 °C | −0.69 °C | ±1.04 °C |
| −25 °C | −0.80 °C | ±1.16 °C |

**Table A1** Measurement uncertainty and temperature corrections for various temperatures at 5 °C steps. The measurement uncertainties are expanded to a coverage of 95 %.

6) Lines 368-370: How does this compare to ice nucleating phyla found in other soil INP studies? Likewise Line 375 for fungi.

Thank you for your comment.
The ASVs that correlate with the INP concentrations are from microbial taxa that were previously unknown to produce ice-nucleation-active proteins. While some of them belong to the same phyla as known INA microorganisms, they do not affiliate to the same genera.

7) Line 126: How many pure water droplets were analysed per experiment? And were the control experiments performed in the same plate as the samples?

Thank you for raising this point. In each plate we included a negative control, which consisted of 64 Milli-Q water droplets. While this is slightly lower compared to 80 droplets analyzed for the samples, the impact on the calculation of INP concentrations is minimal. As described in our methods, the term:

$$\alpha \cdot \ln \left( \frac{\alpha}{\alpha - 1} \right)$$

becomes negligible when $\alpha \gg 1$ making the calculations robust to slight variations in the number of droplets analyzed. Consequently, the use of 64 droplets ($\alpha =1.007895$) vs 80 droplets ($\alpha =1.006303$) does not significantly influence the accuracy or interpretation of the INP data.

8) Line 152: Why is 5% of the droplets freezing used as the onset freezing value?

Thank you for your question. The 5% freezing threshold is used as the onset freezing value to account for the probabilistic nature of ice nucleation experiments. Ice nucleation can be viewed as a probability distribution, where droplets have varying likelihoods of containing an ice-nucleating particle capable of nucleating at a specific temperature. Removing the first and last 5% of the data effectively excludes the tails of this distribution, which represent the extremes. This approach corresponds to a 5% confidence interval on either end, ensuring that the reported onset temperature reflects the more statistically robust and reproducible freezing behavior of the majority of droplets, rather than potential outliers.

9) Line 174: Add the word "respectively" after discussing bacteria and fungi to make clear that the 16S and ITS sequencing refers to specifically to one or the other.

Thank you for your comment. This has been implemented in the new version.

10) Line 181: Please define BSA.

Thank you for your comment. This has been implemented in the new version.

11) Line 184: Missing degree symbols in temperatures.

Thank you for your comment. This has been implemented in the new version.

12) Line 187: What are V3 and V4?

Thank you for your comment. The V3 and V4 regions refer to two variable regions within the 16S rRNA gene, which are widely used in microbial community studies. These regions contain sufficient sequence variability to allow differentiation between bacterial taxa while remaining flanked by conserved regions for primer design. By amplifying these regions, we can achieve high-resolution taxonomic identification of the bacterial communities in our samples. This has been clarified in the revised manuscript.

13) Lines 189-190: The description of the PCR mix is a little confusing. What were the volumes of the components, and what is meant by "2 template DNA" (e.g. should this be 2 ul?) and "2 x KAPA….."?

Thank you for pointing this out. We have clarified the description of the PCR mix in the revised manuscript.

14) Lines 203-204: Why are the products re-quantified after pooling? Due to losses when transferring between vials? Is the total concentration required for the sequencing?

Thank you for your question. The pooled library is re-quantified after pooling to ensure accurate equimolar representation of each sample in the final library. This step is critical because the pool will be multiplexed with other libraries on the Illumina MiSeq platform. Accurate quantification ensures optimal cluster generation during sequencing, preventing over- or under-representation of specific samples and maximizing the quality and consistency of the sequencing output.

15)    Line 214: Define ASVs.

Thank you for your comment. This has been implemented in the new version.

16)    Figures 1 and 2: It would help the reader to color code the locations in Figure 1 with the same colors as used in Figure 2, especially when trying to determine whether there are regional grouping of INP concentrations etc. Ideally the same color coding would be used throughout (e.g. in Supplementary Figures 2, 5 and 6, although it is less important for the Supporting Information compared to the main paper).

We appreciate the suggestion to streamline the color coding for clarity and consistency. We have revised the color scheme to ensure that the locations in Figures 1 and 2, as well as Supplementary Figures 2, 5, and 6, now use the same color coding.

17)    During the Introduction or Results sections, the authors may want to consider the recent work of Herbert et al. using fertile soil representations of INPs rather than simply desert dust: https://egusphere.copernicus.org/preprints/2024/egusphere-2024-1538/

Thank you for this comment and pointing us to this recent research. Since the manuscript mentioned is still in review as a preprint we have decided to leave it out of our manuscript, since we already in our opinion cover relavant litterature.

18)    Line 244: Data for total carbon (TC) is discussed, but this TC data is not shown for the samples (likewise for nitrogen).

Thank you for pointing this out. The data for total carbon (TC) and nitrogen has now been included in Supplementary Table 1, alongside the 16S rRNA gene copy numbers per gram of soil.

19)    Line 251: The sampling of Barry is discussed since they used bulk soil rather than sieved soil, but how was the soil in Tobo and Conen sampled/treated.

Thank you for pointing this out. We have now clarified the sampling and treatment methods used in Conen et al. (2011, 2018) in the manuscript. Both studies prepared their soils by air-drying before sieving to isolate finer fractions. Conen et al. (2011, 2018) used a 63 μm mesh. After sieving, they further separated particles smaller than 5 μm using differential settling techniques. We have added this information to the manuscript to provide a clearer comparison of                                                 the                                                 methodologies.

*"We prepared soil samples for ice nucleation analysis as previously described, with slight modifications (Conen and Yakutin, 2018). The soil samples were placed in a small petri dish and freeze dried overnight (Edwards Micro Modulyo Freeze Dryer). Freeze-drying was chosen*

*instead of air-drying to minimize the potential for microbial activity during the drying process and preserving the original composition of INPs, as prolonged air-drying could induce microbial activity, potentially altering INP concentrations due to production or degradation. The freeze-dried samples were kept in a desiccator to prevent rehydration and subsequently comminuted in a mortar by hand. Mortaring was performed to break down aggregates formed during freeze-drying and ensured effective sieving. Samples were dry sieved with a 125 μm and 63 μm sieve for two minutes using a vibratory sieve shaker (Analysette 3 PRO, Fritsch). The <63 μm fraction was collected for analysis, as this size range represents particles most likely to aerosolize (Fröhlich-Nowoisky et al., 2016)."*

20)  Lines 376-377: "While this phylum is known to be encompass many different lifestyles, only saprotrophic, pathogenic and lichenized fungi are known to produce INPs." – Are there appropriate references that could be used to support this statement?

Thank you, we have now added the following references:
(Pouleur et al., 1992; Fröhlich-Nowoisky et al., 2015; Huffman et al., 2013; Morris et al., 2013).

21)              Line       390:       Consider       citing       Meinander       2022 (https://acp.copernicus.org/articles/22/11889/2022/)            and            Bullard            2016 (https://pubs.usgs.gov/publication/70190769) when discussing emission of dusts and soils from these high Arctic locations.

Thank you for your comment. This has been implemented in the new version.

22)  Figure 5 and caption: Is the concentration in terms of Nm (g-1 of particles) as in Figure 2? If so, please make this clear and have the formatting of parameters/units across the plots be more consistent.

Thank you for noting this. It is indeed in terms of Nm (g-1 of particles), as indicated in Figure 2. We have clarified this in the caption for Figure 5 and ensured that the formatting of parameters and units is consistent across all plots in the manuscript.

23)  Line 440: Could biofouling be another possible mechanism for the loss of proteinaceous material via non-specific adsorption to the membrane material?

Thank you for the idea.
Given that we observe an increase in ice nucleation activity after filtration, it seems less likely that biofouling is the main mechanism responsible for this phenomenon. As mentioned biofouling typically involves the non-specific adsorption of proteins, particles, or other macromolecules to the filter material, which might lead to a decrease in the concentration of active species in the filtrate.
However, if cell lysis were occurring during filtration, this could release cellular contents, including INPs, into the filtrate, potentially leading to an increase in ice nucleation activity.

24)  Line 520: "…while at other locations they are present in solution" – what is meant by this?

Thank you for your comment. The phrase "present in solution" refers to the fact that the remaining INPs in these cases were found in the soluble fraction, indicating that they were likely excreted or detached from microbial membranes.. We have clarified this in the revised manuscript.

*"Additionally, using filtration through a series of filters with decreasing cut-offs, we found that soil INPs at some locations were associated with soil particles or microbial membranes, while at other locations they were present in the soluble fraction, likely excreted or detached from microbial membranes."*

25) Supplementary Figure 1: Is this data for the 63 um sieved samples? Please provide further details about the samples in the caption. Also, provided it does not make the plot too busy, please add the fraction frozen plots for the filtered samples too.

Thank you for your comment regarding Supplementary Figure 1 and for pointing out the need for clarification and additional details. We have addressed your concerns as follows:

1. We have clarified in the caption that the data correspond to soil samples that were pre-sieved to <63 μm before the freezing activity analysis.
2. We have added a detailed explanation of the filtration treatments for each category (<63, Soluble > 1000 kDa, 1000-300 kDa, 300-100 kDa, and <100 kDa).
3. We have added the negative controls (Milliq water samples)

26) Supplementary Figure 2: It would be helpful to the reader to show more sizes on the x-axis, e.g. steps of 20 um or 50 um. The samples all seem to peak at around the 50-60 um region, please note the number in the caption.

Thank you for the suggestion. We have updated Supplementary Figure 2 to show more particle size intervals on the x-axis, now displayed in steps of 20 μm together with gridlines, for improved readability. Additionally, as noted, the samples predominantly peak in the 40–60 μm region, which is now mentioned explicitly in the updated figure caption.

27) Page 6 of the Supporting Information is blank.

Thank you for your comment. This has been corrected in the new version.

28) Supplementary Figure 8: Please include in the caption the definitions of the various parameters shown in the plot.

Thank you for your suggestion. We have updated the caption for Supplementary Figure 8 to include the definitions of the various parameters shown in the plot.

29) There are occasional minor spelling and grammatical errors throughout that can be picked up on a thorough proofread.

Thank you for your comment. We have proofread the manuscript to remove spelling and grammatical errors.

References:

Barry, K. R., Hill, T. C. J., Moore, K. A., Douglas, T. A., Kreidenweis, S. M., DeMott, P. J., and Creamean, J. M.: Persistence and Potential Atmospheric Ramifications of Ice-Nucleating Particles Released from Thawing Permafrost, Environ Sci Technol, 57, 3505-3515, 10.1021/acs.est.2c06530, 2023a.

Barry, K. R., Hill, T. C. J., Nieto-Caballero, M., Douglas, T. A., Kreidenweis, S. M., DeMott, P. J., and Creamean, J. M.: Active thermokarst regions contain rich sources of ice-nucleating particles, Atmospheric Chemistry and Physics, 23, 15783-15793, 10.5194/acp-23-15783-2023, 2023b.

Conen, F. and Yakutin, M. V.: Soils rich in biological ice-nucleating particles abound in ice-nucleating macromolecules likely produced by fungi, Biogeosciences, 15, 4381-4385, 10.5194/bg-15-4381-2018, 2018.

Daily, M. I., Tarn, M. D., Whale, T. F., and Murray, B. J.: An evaluation of the heat test for the ice-nucleating ability of minerals and biological material, Atmospheric Measurement Techniques, 15, 2635-2665, 10.5194/amt-15-2635-2022, 2022.

Fröhlich-Nowoisky, J., Hill, T. C. J., Pummer, B. G., Yordanova, P., Franc, G. D., and Pöschl, U.: Ice nucleation activity in the widespread soil fungus Mortierella alpina, Biogeosciences, 12, 1057-1071, 10.5194/bg-12-1057-2015, 2015.

Fröhlich-Nowoisky, J., Kampf, C. J., Weber, B., Huffman, J. A., Pöhlker, C., Andreae, M. O., Lang-Yona, N., Burrows, S. M., Gunthe, S. S., Elbert, W., Su, H., Hoor, P., Thines, E., Hoffmann, T., Després, V. R., and Pöschl, U.: Bioaerosols in the Earth system: Climate, health, and ecosystem interactions, Atmospheric Research, 182, 346-376, 10.1016/j.atmosres.2016.07.018, 2016.

Huffman, J. A., Prenni, A. J., DeMott, P. J., Pöhlker, C., Mason, R. H., Robinson, N. H., Fröhlich-Nowoisky, J., Tobo, Y., Després, V. R., Garcia, E., Gochis, D. J., Harris, E., Müller-Germann, I., Ruzene, C., Schmer, B., Sinha, B., Day, D. A., Andreae, M. O., Jimenez, J. L., Gallagher, M., Kreidenweis, S. M., Bertram, A. K., and Pöschl, U.: High concentrations of biological aerosol particles and ice nuclei during and after rain, Atmospheric Chemistry and Physics, 13, 6151-6164, 10.5194/acp-13-6151-2013, 2013.

Kunert, A. T., Pöhlker, M. L., Tang, K., Krevert, C. S., Wieder, C., Speth, K. R., Hanson, L. E., Morris, C. E., Schmale Iii, D. G., Pöschl, U., and Fröhlich-Nowoisky, J.: Macromolecular fungal ice nuclei in Fusarium: effects of physical and chemical processing, Biogeosciences, 16, 4647-4659, 10.5194/bg-16-4647-2019, 2019.

Lacher, L., Adams, M. P., Barry, K., Bertozzi, B., Bingemer, H., Boffo, C., Bras, Y., Büttner, N., Castarede, D., Cziczo, D. J., DeMott, P. J., Fösig, R., Goodell, M., Höhler, K., Hill, T. C. J., Jentzsch, C., Ladino, L. A., Levin, E. J. T., Mertes, S., Möhler, O., Moore, K. A., Murray, B. J., Nadolny, J., Pfeuffer, T., Picard, D., Ramírez-Romero, C., Ribeiro, M., Richter, S., Schrod, J., Sellegri, K., Stratmann, F., Swanson, B. E., Thomson, E. S., Wex, H., Wolf, M. J., and Freney, E.: The Puy de Dôme ICe Nucleation Intercomparison Campaign (PICNIC): comparison between online and offline methods in ambient air, Atmospheric Chemistry and Physics, 24, 2651-2678, 10.5194/acp-24-2651-2024, 2024.

Morris, C. E., Sands, D. C., Glaux, C., Samsatly, J., Asaad, S., Moukahel, A. R., Gonçalves, F. L. T., and Bigg, E. K.: Urediospores of rust fungi are ice nucleation active at > −10 °C and harbor ice nucleation active bacteria, Atmospheric Chemistry and Physics, 13, 4223-4233, 10.5194/acp-13-4223-2013, 2013.

Polen, M., Brubaker, T., Somers, J., and Sullivan, R. C.: Cleaning up our water: reducing interferences from nonhomogeneous freezing of "pure" water in droplet freezing assays of ice-nucleating particles, Atmos. Meas. Tech., 11, 5315-5334, 10.5194/amt-11-5315-2018, 2018.

Pouleur, S., Richard, C., Martin, J. G., and Antoun, H.: Ice Nucleation Activity in Fusarium acuminatum and Fusarium avenaceum, Appl Environ Microbiol, 58, 2960-2964, 10.1128/aem.58.9.2960-2964.1992, 1992.

Schwidetzky, R., de Almeida Ribeiro, I., Bothen, N., Backes, A. T., DeVries, A. L., Bonn, M., Frohlich-Nowoisky, J., Molinero, V., and Meister, K.: Functional aggregation of cell-free proteins enables fungal ice nucleation, Proc Natl Acad Sci U S A, 120, e2303243120, 10.1073/pnas.2303243120, 2023.

Wieber, C., Rosenhøj Jeppesen, M., Finster, K., Melvad, C., and Šantl-Temkiv, T.: Micro-PINGUIN: microtiter-plate-based instrument for ice nucleation detection in gallium with an infrared camera, Atmospheric Measurement Techniques, 17, 2707-2719, 10.5194/amt-17-2707-2024, 2024.

---

## Author Comment (AC4)

Comment by the topic editor
The three reviewers indicated that the manuscript is suitable for publication in Aerosol Research subject to some revisions. The authors should answer the reviewers' comments/suggestions before a revised manuscript can be considered for final publication. In that case, the revised manuscript will be sent for another round of reviews by the same reviewers.

The topic editor further suggests modifying lines 128 to 132 in the manuscript by a larger explanation. Also, the authors should note that term $\alpha \ln [(\alpha/(1-\alpha)]$ does not become negligible (as written in the manuscript), but tends to unity when $\alpha$ is very large.

Suggested change instead of lines 128 to 132:
The mean number of INPs per volume in the sample that are active at a given temperature can be calculated assuming that the locations of these INPs in the sample volume are statistically independent. Then, the probability of having a given number of droplets without INPs (that is, the fraction of no frozen droplets) is given by the binomial distribution, and the remaining fraction of droplets containing INPs (fraction of frozen droplets) leads to the relation:
Their Eq. (1)
Where $f$ is the frozen fraction (i.e., the fraction of frozen droplets) and $V$ is the droplet volume in each 130 vial of the assay and $\alpha$ is the number of droplets per sample (80 in this study).
Given that $\alpha >> 1$ in our study, the term $\alpha \ln [(\alpha/(1-\alpha)]$ is close to unity. Therefore Eq. (1) simplifies to Eq (2)

We thank the editor for this clarification and suggestion.
We have now incorporated the suggested change into the revised manuscript:

*"The mean number of INPs per volume in the sample that are active at a given temperature can be calculated assuming that the locations of these INPs in the sample volume are statistically independent. Then, the probability of having a given number of droplets without INPs (that is, the fraction of no frozen droplets) is given by the binomial distribution, and the remaining fraction of droplets containing INPs (fraction of frozen droplets) leads to the following general Eq. (1):*

$$N_v = \frac{-\ln(1-f)}{V \cdot \alpha \cdot \ln\left(\frac{\alpha}{\alpha-1}\right)},$$

*(1)*

*Where $f$ is the frozen fraction (i.e., the fraction of frozen droplets) and $V$ is the droplet volume in each vial of the assay and $\alpha$ is the number of droplets per sample (80 in this study).*
*Given that $\alpha >> 1$ in our study, the term $\alpha \cdot \ln\left(\frac{\alpha}{\alpha-1}\right)$ is close to unity. Therefore Eq. (1) simplifies to Eq (2)*

$$N_v = \frac{-\ln(1-f)}{V},$$

*(2)"*

Subsequently this simplified form was used for the calculations in this study (Vali, 1971).